# DEFENDING AGAINST BACKDOOR ATTACKS USING ENSEMBLES OF WEAK LEARNERS

## ABSTRACT

A recent line of work has shown that deep networks are susceptible to backdoor data poisoning attacks. Specifically, by injecting a small amount of malicious data into the training distribution, an adversary gains the ability to control the behavior of the model during inference. We propose an iterative training procedure for removing poisoned data from the training set. Our approach consists of two steps. We first train an ensemble of weak learners to automatically discover distinct subpopulations in the training set. We then leverage a boosting framework to exclude the poisoned data and recover the clean data. Our algorithm is based on a novel bootstrapped measure of generalization, which provably separates the clean from the dirty data under mild assumptions. Empirically, our method successfully defends against a state-of-the-art dirty label backdoor attack. We find that our approach significantly outperforms previous defenses.

## 1 OVERVIEW

The past few years has seen the rapid adoption of deep learning in real world applications, from digital personal assistants to autonomous vehicles. This trend shows no sign of abating, given the remarkable (super)human performance of deep neural networks on tasks such as computer vision (He et al., 2016a), speed recognition (Graves et al., 2013), and game playing (Silver et al., 2016).

However, this widespread integration of deep networks presents a potential security risk, particularly in performance- and safety-critical applications. In this work, we focus on defending against *backdoor attacks* (Chen et al., 2017; Adi et al., 2018). Specifically, it has been demonstrated that deep networks can be attacked by injecting small amounts of poisoned (i.e., maliciously perturbed) data during training to create a backdoor in the model; once installed, an adversary can exploit the backdoor to change the network's predictions at inference time. For instance, Gu et al. (2019) demonstrate a backdoor that causes a model to misclassify stop signs as speed signs by applying a (physical) sticker. These attacks are particularly pernicious in that the accuracy of the model on unperturbed data is generally not affected by the backdoor, thus making it difficult to identify compromised models during standard operation.

**Techniques** We first introduce the notion of **self-expanding sets**, which based on a bootstrapped measure of how well a set generalizes to itself. Under certain compatibility properties, we show that the process of identifying self-expanding sets naturally separates a dataset into a collection of homogeneous components (i.e., completely clean or completely poisoned subsets of the training data). Given such a collection, we then provide a method to identify the clean distribution by boosting an ensemble of weak learners over the components.

To separate the training set into homogeneous components, we present the **Inverse Self-Paced Learning** algorithm. This algorithm uses quantile statistics to repeatedly identify and exclude samples with high loss. Recursively applying the technique to sets of excluded samples produces the collection of homogeneous components. We prove sufficient conditions for the convergence of the algorithm.

**Experimental Evaluation** We implement the proposed Inverse Self-Paced Learning algorithm within our boosting framework and evaluate its performance on two different backdoor attacks on CIFAR-10 (**?**). Our method completely defends against the attacks in almost every setting and

substantially reduces the success rate of the attack in the remaining cases, while reducing the accuracy on clean data by only 2-3%. Our results also show that previous approaches (Tran et al., 2018; Chen et al., 2018; Shen and Sanghavi, 2019) are substantial weaker at identifying poisoned samples. These previous approaches also either require an explicit upper bound on expected amount of poison or suffer from high levels of false positives. Our approach thus presents a novel, empirically verified method for defending against backdoor attacks.

## 2 RELATED WORK

Our separation results can be viewed as solutions to a clustering problem, where we exploit weak supervision in the form of class labels, related via our notions of self-expansion and compatibility. Many previous works analyze similar properties for weak or semi-supervised learning based on clustering (Seeger, 2000; Rigollet, 2007; Singh et al., 2008) or expansion (Wei et al., 2020). In general, these works define expansion as an intrinsic property of the input data, rather than with respect to a (weak) learner as we do. Balcan et al. (2005) show that under a similar expansion property, learners fit independently to two different "views" of the data can supervise each other to improve their joint performance. However, a major departure from these prior works is that we do not use an expansion property to leverage a small set of trusted or confident labels for minimizing a global classification error, but rather use self-expansion to identify homogeneous components by fitting weak learners to certain local minima.

We also introduce the Inverse Self Paced Learning algorithm for efficiently finding self-expanding sets. Self paced learning (SPL) was introduced by Kumar et al. (2010) as an type of curriculum learning (Bengio et al., 2009). SPL is a heuristic that dynamically creates a curriculum based on the losses of the model after each epoch, so as to incorporate the easier samples first. SPL and its variants have been observed to be resilient to noise both in theory (Meng et al., 2016) and practice (Jiang et al., 2018; Zhang et al., 2020), though prior works focus mostly on clean accuracy under unrealizeable label noise. In contrast, we measure targeted misclassification accuracy under more challenging noise distributions that are adversarially selected to be realizeable.

Finally, several prior works propose methods for defending against backdoor attacks on neural networks. In general, it has been observed that standard techniques from robust statistics applied directly to the data do not successfully identify the poisoned data (Tran et al., 2018). The standard approach is therefore to first fit a deep network to the poisoned distribution, then apply techniques to the learned representations in the network layers. The Activation Clustering defense (Chen et al., 2018) uses dimensionality reduction followed by k-means clustering (k=2) on the activation patterns, and discards the smallest cluster. Tran et al. (2018) propose a Spectral Signature defense that removes the data with the top $\epsilon$ eigenvalues, where $\epsilon$ is set to 1.5 times the amount of expected poison. TRIM (Jagielski et al., 2021) (for linear regression) and Iterative Trimmed Loss Minimization (Shen and Sanghavi, 2019) (for generalized linear models and deep neural networks) iteratively train on a subset of the data after removing a constant fraction of the samples with the highest loss. However, the majority of works do not evaluate their defenses on CIFAR-10, opting instead for simpler datasets, such as traffic signs or MNIST. Furthermore the triggers skew large and obvious (such as 3x3 patches (Qiao et al., 2019) or legible text overlays (Gao et al., 2019a)), and are often constrained to lie in the center of the image; our evaluation shows that existing defenses fail when evaluated on the more subtle triggers we use.

## 3 BACKGROUND AND SETTING

We first establish some basic notation and the scope of our classification setting. Let $X$ be the input space, $Y$ be the label space, and $L(\cdot, \cdot)$ be a loss function over $Y \times Y$. We assume a bounded loss function, which includes many commonly used loss functions such as the zero-one or cross entropy loss. Given a target distribution $\mathcal{D}$ supported on $X \times Y$ and a parametric family of functions $f_\theta$, the goal is to find the parameters $\theta$ that minimize the population risk $R(\theta) := \int_{X \times Y} L(f_\theta(x), y) dP_{\mathcal{D}}(x, y)$.

The learning problem $(f_\theta, \mathcal{D})$ is realizable if (1) for every label $y$, the marginals $\mathcal{D}(\cdot|y)$ have disjoint support; and (2) there exist ground truth parameters $\theta^*$ with $R(\theta^*) = 0$. For simplicity, we assume a (possibly stochastic) learning algorithm $\mathcal{A}$ that performs empirical risk minimization, i.e., given a set of samples $T$, $\mathcal{A}$ tries to return $\theta$ minimizing $R_{emp}(\theta; T) := \sum_{i \in T} L(f_\theta(x_i), y_i)$. Clearly, given

enough training samples $S = \{(x_1, y_1)...,(x_n, y_n)\}$ iid from $\mathcal{D}$, the empirical risk gets arbitrarily close to the population risk. We will identify the training set $S$ with its indices $[n]$.

## 3.1 DATA AND THREAT MODEL

We consider a mixture of $n$ distributions $\{(\alpha_i, \mathcal{D}_i)\}_{i=1}^n$ such that $\mathcal{D} = \cup_i \{\mathcal{D}_i\}$ and $\sum_i \alpha_i = 1$. We observe $N$ inputs according to the following two-step procedure:

$$d \sim \text{Cat}(\alpha_1, ..., \alpha_n) \tag{1}$$
$$x, y \sim \mathcal{D}_d \tag{2}$$

where $\text{Cat}(\cdot)$ is a categorical random variable that returns $i$ with probability $\alpha_i$. If $S$ is a set of samples produced by this process, for any subset $S' \subseteq S$, we will denote the samples drawn from the $i^{th}$ distribution as $S'_i$, so that $S' = \cup_i S'_i$.

Our evaluation focuses on the backdoor data poisoning model. It has been observed empirically that injecting a small amount of malicious data into the training distribution effectively installs a backdoor in the model, whereby the behavior on clean data is otherwise unaffected, but an attacker can cause targeted misclassification during inference by overlaying a small trigger. In this case sampling from the training distribution is modeled as follows:

$$d \sim \text{Cat}(\alpha_1, ..., \alpha_n) \tag{3}$$
$$x, y \sim \mathcal{D}_d \tag{4}$$
$$p \sim \text{Bern}(\rho) \tag{5}$$
$$x \leftarrow \tau(x), y \leftarrow \pi(y), \text{ if } p \tag{6}$$

where $\tau(\cdot)$ is the function which applies a small trigger, $\pi(\cdot)$ is a permutation on classes, and $\rho$ controls the probability of observing a poisoned sample. Note that this procedure can easily be replicated within the original data model. We will also assume the attack is non-trivial in the sense that the perturbed source distributions $\tau(\mathcal{D}_i)$ and target distributions $\mathcal{D}_{\pi(i)}$ are disjoint for all $i$.

Given a model $f_\theta(\cdot)$, we measure the success of the attack using the *targeted misclassification rate*:

$$A_{emp}(\theta; T) := \sum_{i \in T} [f_\theta(x_i) = y_i \wedge f_\theta(\tau(x_i)) = \pi(y_i)] \tag{7}$$

In other words, the attack succeeds if, during inference, it can flip the label of a correctly-classified instance by applying the trigger $\tau(\cdot)$.

## 3.2 LEARNING OBJECTIVE

We formulate our learning objective in the general data model of Equations 1-2. Without loss of generality, we assume the first $p$ distributions are *primary distributions*, and the remaining $n - p$ are *noise distributions*. Given a training set $S$, we write $S_P = S_1 \cup ... \cup S_p$ for the samples from the primary distributions, and $S_N = S_{p+1} \cup ... \cup S_n$ for the samples from the noise distributions.

Our goal is to learn parameters $\tilde{\theta}$ which correspond to training only on the primary distributions: $\tilde{\theta} := \mathcal{A}(S_P)$. Note this objective differs significantly from simply minimizing the risk over the primary distributions when the mixed distribution $\mathcal{D}$ is realizable, i.e., we are also interested in avoiding effects that occur on portions of the input space that have low density in the primary distributions. More explicitly, in terms of the data poisoning threat model (Equation 7), we note that training on $S_P \cup S_N$ would yield low risk over the unperturbed distributions, but also high targeted misclassification risk. Conversely, for sufficiently separated distributions $\tau(\mathcal{D}_i)$ and $\mathcal{D}_{\pi(i)}$, we expect that the hypothesis class $f_\theta$ enjoys low targeted misclassification risk when trained only on clean data.

## 4 SEPARATION OF MIXED DISTRIBUTIONS

We next introduce the main theoretical properties that allow us to separate the primary and noise distributions. Our main tool is a property of "self-expanding" sets; intuitively, given a set, we

resample at a given rate and measure how well the learning algorithm generalizes to the rest of the set. Given primary and noise components satisfying certain compatibility properties, we show that the set with the optimal expansion must be homogeneous, i.e., drawn entirely from either the primary or noise components. Finally, we fit weak learners to the recovered (homogeneous) sets in a simplified boosting framework to identify the primary component.

## 4.1 Self-expansion and compatibility

We begin by stating a formal characterization of the self-expanding property of sets:

**Definition 4.1** (Self-expansion of sets.). *Let $S$ and $T$ be sets. We define the $\alpha$-expansion error of $S$ given $T$ for all $0 \leq \alpha \leq 1$ such that $\alpha|S|$ is integral as*

$$\epsilon(S|T; \alpha) := |S|^{-1} \mathbb{E}[R_{emp}(\mathcal{A}(S' \cup T); S)] \tag{8}$$

*where the expectation is over both the randomness in $\mathcal{A}$ and $S'$, a random variable of $\alpha|S|$ samples drawn from $S$ with replacement.*

This self-expansion property measures the ability of the learning algorithm $\mathcal{A}$ to generalize to the empirical distribution of a set $S$ with the help of additional training samples $T$; intuitively, a smaller expansion error means that the set $S$ is both "easier" and "more homogeneous" with respect to the learning algorithm and $T$. When $T = \emptyset$ we will also write $\epsilon(S; \alpha)$ instead of $\epsilon(S|\emptyset; \alpha)$. $\alpha$ is also referred to as the *subsampling rate*. Finally, we will extend $\epsilon(S|T; \alpha)$ to all $0 \leq \alpha \leq 1$ by linearly interpolating between the value at integral sample sizes.

We now use self-expansion to define a notion of compatibility between sets:

**Definition 4.2** (Compatibility of sets.). *A (nonempty) set $T$ is $\alpha$-compatible with set $S$ with margin $\delta \geq 0$ if*

$$\epsilon(S|T; \alpha) + \delta \leq \epsilon(S; \alpha) \tag{9}$$

*where the expectation is over the same random variables as in the definition of self-expansion. Furthermore, $T$ is completely $\alpha$-compatible with $S$ if all (nonempty) subsets $T' \subseteq T$ are $\alpha$-compatible with $S$. Conversely, $T$ is $\alpha$-incompatible with $S$ if the opposite holds, i.e.,*

$$\epsilon(S; \alpha) + \delta \leq \epsilon(S|T; \alpha) \tag{10}$$

*(and similarly for complete incompatibility). We also say that strict compatibility (incompatibility) holds when $\delta > 0$.*

In other words, $T$ is compatible with $S$ if the self-expansion error of $S$ given $T$ is not worse than the self-expansion error of $S$ by itself.

In what follows, we make use of the following assumptions about expansion:

**Assumption 4.3** (Properties of expansion). *The learning procedure satisfies the following:*

*(1) $\epsilon(S|T; \alpha)$ is a convex function of $\alpha \in [0, 1]$ such that $\epsilon(S|T; 1) = 0$ for all $S, T$*

*(2) if $T$ is $\alpha$-incompatible with $S$, then $T$ is $\beta$-incompatible with $S$ for all $\beta \geq \alpha$.*

The first assumption rules out the existence of pathological sets where increasing the number of samples degrades performance and also says that memorization of the training set always occurs. Convexity holds when the expected marginal information gained from additional samples decreases as the number of samples increases. The second assumption says that increase the amount of data from $S$ in the training set (which always improves performance, regardless of the compatibility of $T$) should not flip an incompatible $T$ into a compatible set.

The following key property enables us to separate the primary and noise distributions. Intuitively, we want the primary and noise mixture components to be negatively correlated (or at least independent) in the sense that training on a noise distribution should not improve performance on a primary distribution, and vice versa:

**Property 4.4** (Incompatibility of primary and noise distributions). *Let $\alpha$ be given. Then any pair of (nonempty) sets $S_P$ and $S_N$ drawn from $\mathcal{D}_1 \cup ... \cup \mathcal{D}_p$ and $\mathcal{D}_{p+1} \cup ... \cup \mathcal{D}_n$, respectively, are strictly and completely $\alpha$-incompatible.*

Our technique is designed for separating primary and noise distributions that satisfy this property.

We are now ready to state the main result of this section. Given Property 4.4, we show that any subset of $S$ which achieves the minimum expansion error consists entirely of data drawn from either the primary or noise distributions:

**Theorem 4.5** (Sets minimizing expansion error are homogeneous.). *Let $S = S_1 \cup ... \cup S_n$ be a set of samples drawn from a mixture of distributions $\{(\alpha_i, \mathcal{D}_i)\}_{i=1}^n$. Define*

$$S^* := \arg \min_{S' \subseteq S} \epsilon(S'; \alpha) \tag{11}$$

*for some expansion factor $\alpha$. Then if Property 4.4 holds for $\alpha$, we have either that $S^* \subseteq S_P$ or $S^* \subseteq S_N$, where $S_P = S_1 \cup ... \cup S_p$ and $S_N = S_{p+1} \cup ... \cup S_n$.*

We defer the proof to Appendix A. Intuitively, if two distributions are incompatible, then adding data from one distribution to a homogeneous set of the other should only increase the self-expansion error.

**Remark 1**  Theorem 4.5 relies crucially on the incompatibility between the samples from the primary distribution $S_1 \cup ... \cup S_p$ and the noise distribution $S_{p+1} \cup ... \cup S_n$ derive the homogeneity of $S^*$, a condition which depends on the interaction between the data $S$ and the learning algorithm $\mathcal{A}$. We note that the requirement is empirically satisfied in many data poisoning settings. For example, a common adversary for backdoor attacks against deep neural networks inserts a small synthetic patch in the corner of the image, which, by design, is a location on which the classification does not depend. In this case, the labels for the primary and noise distributions depend on disjoint dimensions of the input, which gives a very clean example of incompatible distributions; given the number of shared (spurious) features, empirical results suggest that the two distributions are, in fact, strictly incompatible for moderately large sets as well.

**Remark 2**  In general, the larger $\alpha$ is in Property 4.4, the easier it is to estimate the value of $\epsilon(S; \alpha)$; the most convenient case would be for incompatibility to hold even when $\alpha = 1$, in which case $\epsilon(S; \alpha)$ can be evaluated exactly with one call to $\mathcal{A}$. Unfortunately, for overparameterized models trained using empirical risk minimization, we have that $\epsilon(S; 1) = 0$ for all $S$ (since we assume the problem is realizable in the limit). One method to circumvent this problem is to prevent $\mathcal{A}$ from converging, e.g., by using early stopping. In fact, it is well known that regularizing deep neural networks trained with Stochastic Gradient Descent using early stopping is resilient to noise (Li et al., 2020). In our experiments, we find that combining early stopping with our self-expansion property leads to further improvements in performance.

## 4.2 IDENTIFICATION USING WEAK LEARNERS

The development of the previous section suggests an iterative approach to separating the primary and noise distributions. In particular, if we fix the expansion factor $\alpha$, at each step, we can identify the set $S^*$ which achieves the lowest expansion error and remove it from the training set. Repeating this procedure partitions the training set into groups of compatible sets. While this suffices to *separate* the primary and noise distributions, it remains to *identify* which components belong to the primary distribution.

We next propose a simplified boosting framework for identification of the primary distribution. We assume the setting of binary classification and use the 0-1 loss, so that the empirical risk simply counts the number of elements which are misclassified. Our approach is to fit a weak learner to each component, then use each learner to vote on the other components.

Algorithm 1 presents our approach for boosting from homogeneous sets. The subroutine $\text{Loss}_{0,1}$ takes a set of parameters and a set of samples, and returns the empirical zero-one loss over the entire set. Note that votes are weighted by size.

The correctness of Algorithm 1 follows from an analogous compatibility property (cf. Property 4.4):

**Property 4.6** (Compatibility of primary distribution). *Let $\alpha$ be given. Then any pair of (nonempty) sets $S_i$ and $S_j$ drawn from $\mathcal{D}_i$ and $\mathcal{D}_j$, respectively, such that $i, j \le n$, are strictly and completely $\alpha$-compatible.*

Finally, we also require unbiased priors for weak learners:

---

**Algorithm 1** Boosting Homogeneous Sets

---

**Input:** Homogeneous sets $S_1, ..., S_N$, total number of samples $n$, number of estimates $B$, weak learner $\mathcal{A}$
**Output:** Votes $V_1, ..., V_N$
 1: $C_1, ..., C_N \leftarrow 0$
 2: **for** $i = 1$ **to** $N$ **do**
 3:    $V_{i1}, ..., V_{iN} \leftarrow 0$
 4:    **for** $j = 1$ **to** $B$ **do**
 5:        $\theta_{ij} \leftarrow \mathcal{A}(S_i)$
 6:        **for** $k = 1$ **to** $N$ **do**
 7:            $V_{ik} \leftarrow V_{ik} + \text{Loss}_{0,1}(\theta_{ij}; S_k)/B$
 8:        **end for**
 9:    **end for**
10:    **for** $k = 1$ **to** $N$ **do**
11:        **if** $V_{ik} > |S_k|/2$ **then**
12:            $C_k \leftarrow C_k + |S_i|$
13:        **end if**
14:    **end for**
15: **end for**
16: **for** $i = 1$ **to** $N$ **do**
17:    $V_i \leftarrow C_i > n/2$
18: **end for**

---

**Property 4.7** (Weak learners are unbiased.). *Let $S_i$ be any sets. Then for any untrained weak learner, we have also that $\mathbb{E}[R_{emp}(\mathcal{A}(\emptyset); S_i)] = |S_i|/2$.*

This condition is necessary in that the weak learners should not be biased toward learning the noise distributions. Finally, we state the main result of this section, whose proof is deferred to Appendix A.

**Theorem 4.8** (Identification of primary samples). *Let $S$ be a set of samples drawn from a mixture of distributions $\{(\alpha_i, \mathcal{D}_i)\}_{i=1}^{n}$ such that Properties 4.4, 4.6, and 4.7 hold, and assume that the ratio of primary samples $p = |S_P|/|S| > 1/2$. Let $S_1, ..., S_N$ be a partition of $S$ produced by iteratively applying Theorem 4.5. Then if $\mathcal{A}$ is deterministic, Algorithm 1 returns $1$ with $B = 1$ for all components containing samples from the primary distribution, and $0$ otherwise.*

*If $\mathcal{A}$ is stochastic, the same result holds with probability*

$$[1 - 2\exp(-2\delta^2 B)]^{MN} \tag{12}$$

*where $M$ is the number of primary components, and $B$ is the number of independent weak learners used to fit each primary component.*

**Remark 3**    At a high level, the approach to identifying the primary distribution presented in Theorems 4.5 and 4.8 follows a simplified boosting framework: at each step, we fit a weak learner to a subset of the distribution, then reweight the remaining training samples by removing the identified component; the ensemble of weak learners is then aggregated using a majority vote. However, our setting is somewhat unique so for clarity we mention several key differences. First, in general the objective of standard boosting is to achieve low population risk, thus the reweighting is performed via more sophisticated methods such as using the empirical loss of the ensemble thus far, e.g., AdaBoost (Freund et al., 1996); in contrast, in our setting there are subpopulations over which we would actually like to maximize the risk. Another difference is that in standard boosting, the ensemble is used during inference to vote on new observations to perform classification, whereas in our algorithm, we use each learner to vote over components of the training set to filter out the noise distributions. Finally, note that we can succeed with arbitrary probability by taking the number of samples $B$ to infinity.

## 5    Inverse self-paced learning

A major question raised by Theorem 4.5 is how to identify the set $S^*$ in its statement. In this section, we propose an algorithm called *Inverse Self-Paced Learning* (ISPL) to solve this problem. Rather

---

**Algorithm 2** Inverse Self-Paced Learning

---

**Input:** training set $S$, total iterations $N$, annealing schedule $1 \geq \beta_0 \geq ... \geq \beta_N = \beta_{\min} > 0$, expansion $\alpha \leq 1$, momentum $\eta$, incremental learning procedure $\mathcal{A}$, initial parameters $\theta_0$
**Output:** $S_N \subseteq S$ such that $|S_N| = \beta_N|S|$
1: $S_0 \leftarrow S$
2: $L \leftarrow \mathbf{0}$
3: **for** $t = 1$ **to** $N$ **do**
4:     $S' \leftarrow \text{Sample}(S_{t-1}, \alpha)$
5:     $\theta_t \leftarrow \mathcal{A}(S', \theta_{t-1})$
6:     $L \leftarrow \eta L + (1 - \eta)R_{emp}(\theta_t; S)$
7:     $S_t \leftarrow \text{Trim}(L, \beta_t)$
8: **end for**

---

than optimizing over all possible subsets of the training data, our objective will instead be to minimize the expansion error over subsets of fixed size $\beta|S|$. The optimization objective is defined as:

$$S_\beta^* := \arg\min_{S' \subseteq S : |S'| = \beta|S|} \epsilon(S'; \alpha) \tag{13}$$

We attempt to solve for $S_\beta^*$ by alternating between optimizing parameters $\theta_t$ and the training subset $S_t$. More explicitly, given $S'$ we update $\theta$ using a single subset from $S'$ of size $\alpha^{-1}|S'|$. Then we use $\theta$ to compute the loss for each element in $S$, and set $S'$ to be the $\beta$ fraction of the samples with the lowest losses. To encourage stability of the learning algorithm, the losses are smoothed with an optional momentum term $\eta$. We also anneal the parameter $\beta$ from an initial value $\beta_0$ down to the target value $\beta_{\min}$ in order encourage more global exploration in the initial stages.

Algorithm 2 presents the full algorithm. In addition to the incremental learning procedure $\mathcal{A}$ (e.g., standard SGD), the subroutine Sample takes a training set $S$ and returns $\alpha|S|$ elements uniformly at random; while Trim takes losses $L$ and returns the $\beta|L|$ samples with the lowest loss.

Finally, we show for certain parameters that Algorithm 2 converges on the following objective over the training set $S$:

$$F(\theta_t, v_t; \beta_t) := \sum_{i \in S} v_t[i]L(f_{\theta_t}(x_i), y_i) + c\max(0, \beta_t|S| - |v_t|) \tag{14}$$

where $v_t$ is a 0-1 vector, $\beta_t$ is decreasing, and $L(\cdot, \cdot) \leq c$.

**Proposition 5.1.** *Let $\alpha = 1$ and $\eta = 0$ in the setting of Algorithm 2, and assume that $\mathcal{A}$ returns the empirical risk minimizer. Then we have that for each round of the algorithm, $F(\theta_t, v_t; \beta_t)$ is decreasing in $t$ and furthermore, $|F(\theta_t, v_t; \beta_t) - F(\theta_{t+1}, v_{t+1}; \beta_{t+1})| \xrightarrow{t \to \infty} 0$.*

We defer the proof of Proposition 5.1 to Appendix A.

**Remark 4** When $\alpha = 1$ and $\eta = 0$, we recover vanilla self-paced learning (SPL) with two major differences. First, we start on the full set of samples and train on incrementally smaller sets, while SPL starts with a small set of samples and trains on larger sets. This discrepancy is due to the differing objectives; whereas SPL is a heuristic for converging faster to a global minimizer of the population loss by training first on easy samples, ISPL attempts to converge to a local minimum over a subpopulation. Second, our annealing schedule is defined using the quantile statistics, while SPL uses an absolute loss threshold that generally scales by a multiplicative factor in each iteration. We chose this to counteract the propensity of deep neural networks to suddenly and rapidly interpolate the training data; in our experiments, we found this behavior made the performance of ISPL very sensitive to the specific annealing schedule when using absolute losses. Conversely, in SPL the final threshold is generally set high enough that most (or all) the samples are incorporated by the end, and so the specific schedule may have a smaller impact on the final performance.

## 6 EXPERIMENTAL EVALUATION

We evaluate our defense against the standard patch-based backdoor attack with dirty labels, where the adversary inserts a small patch into a training image from the source class, then changes the label

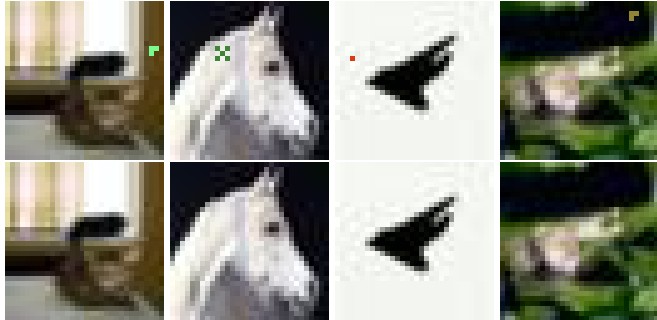

Figure 1: CIFAR-10 images with triggers applied (top), selected to maximize trigger visibility.

Table 1: Results against dirty label backdoor adversary for select pairs of CIFAR-10 classes using a single pixel trigger. The numbers in column 1 refer to the standard CIFAR-10 labels (e.g., 0 = airplane, 1 = automobile, etc.). Column 2 gives the (x, y) coordinates of the trigger. S = source class, T = target class, C = clean accuracy (higher is better), A = targeted misclassification rate (lower is better). Results for our method are in the last two columns under TW (this work).

| S / T | pos | $\epsilon$ | No defense | | SS | | AC | | ITLM | | TW | |
|---|---|---|---|---|---|---|---|---|---|---|---|---|
| | | | C | A | C | A | C | A | C | A | C | A |
| 2 / 5 | (27, 9) | 5 | 94.5 | 75.6 | 94.3 | 74.7 | 92.4 | 53.9 | 94.7 | 79.4 | 92.5 | 0.1 |
| | | 10 | 94.6 | 95.2 | 94.4 | 0.0 | 92.9 | 81.8 | 94.7 | 92.4 | 93.0 | 0.0 |
| | | 20 | 94.7 | 98.1 | 94.2 | 0.0 | 92.6 | 89.4 | 94.6 | 96.3 | 92.8 | 0.0 |
| 1 / 3 | (15, 4) | 5 | 94.8 | 99.3 | 94.6 | 50.0 | 92.2 | 39.2 | 94.6 | 57.2 | 92.0 | 0.1 |
| | | 10 | 94.5 | 99.2 | 94.2 | 10.4 | 92.0 | 47.9 | 94.5 | 75.0 | 92.9 | 0.3 |
| | | 20 | 94.5 | 98.8 | 94.3 | 1.5 | 91.9 | 60.5 | 94.2 | 92.3 | 92.3 | 1.3 |
| 8 / 6 | (4, 1) | 5 | 94.7 | 84.1 | 94.8 | 80.5 | 92.8 | 73.3 | 94.4 | 76.3 | 93.1 | 0.0 |
| | | 10 | 94.6 | 96.4 | 94.2 | 96.0 | 92.2 | 97.0 | 94.5 | 94.2 | 93.0 | 0.0 |
| | | 20 | 94.1 | 98.1 | 94.3 | 0.0 | 92.5 | 96.4 | 94.1 | 96.4 | 92.9 | 0.0 |
| 9 / 2 | (4, 27) | 5 | 94.9 | 98.0 | 94.4 | 65.7 | 92.8 | 79.7 | 94.7 | 97.6 | 93.0 | 0.0 |
| | | 10 | 94.9 | 99.1 | 94.6 | 0.0 | 92.6 | 80.8 | 94.4 | 99.3 | 92.9 | 0.0 |
| | | 20 | 94.7 | 99.1 | 94.3 | 0.0 | 93.1 | 98.9 | 94.1 | 99.1 | 93.1 | 0.0 |

of the image to the target class. The goal is to induce the learner to misclassify images from the source class as the target class upon application of the patch. Results in this section use a standard PreActResNet18 architecture (He et al., 2016b) that achieves 94% accuracy on CIFAR-10 when trained on a clean dataset. The appendix provides full experimental details and additional results.

## 6.1 RESULTS

Our implementation of the dirty label backdoor adversary follows the threat model described in Gu et al. (2017). The perturbation function $\tau$ simply overlays a small pattern on the image. For evaluation, we use the same dataset (CIFAR-10 (?)) and setup for our experiments as Tran et al. (2018). Example pairs of clean and poisoned data are shown in Figure 1.

Table 1 presents results for the single-pixel backdoor attack, in which the adversary randomly selects a position and color for the backdoor and applies the trigger by replacing the pixel at that position with the selected color. The first column, S / T, presents numbers in the form S / T, where S is the source class and T is the target class in CIFAR-10. The goal of the attacker is to induce the network to misclassify poisoned images from the S class to the T class. The second column, pos, presents numbers in the form (X,Y) where X,Y is the position of the single pixel trigger. The third column, $\epsilon$, presents the percentage of the source class in the training set that is poisoned by the adversary.

We report results for our defense, This Work (TW), in the last column, in addition to four baseline defenses: 1) No defense, 2) Spectral Signatures (SS) (Tran et al., 2018), 3) Activation Clustering (AC) (Chen et al., 2018), and 4) Iterative Trimmed Loss Minimization (ITLM) (Shen and Sanghavi, 2019). For each defense we report the percent accuracy over the clean images in the test set (column C, higher is better, maximum is 100% when all clean images are classified correctly) and the targeted misclassification rate (Equation 7) over patched images of the target class in the test set (column A, lower is better, minimum is 0% when none of the poisoned images are misclassified). The results show that the technique we present in this paper 1) almost completely defends against this attack (column A ranges from 0.0% to 1.3%) at the cost of 2) a small (roughly 2%) decrease in the clean accuracy (column C, clean accuracies around 92-93%). All other defenses exhibit significant vulnerability to this attack (column A, No defense, SS, AC, and ITLM).

## 6.2 DISCUSSION

Many data poisoning defenses in the literature are evaluated on simpler datasets than those considered in this paper, such as traffic signs (GTSRB (Houben et al., 2013) or LISA (Mogelmose et al., 2012)) and MNIST (**?**). Furthermore, these datasets are tested in conjunction with larger or otherwise more obvious triggers. For instance, the Neural Cleanse (Wang et al., 2019) defense uses a 4x4 white box as the trigger on the MNIST and GTSRB datasets; MESA (Qiao et al., 2019) uses a 3x3 image as the trigger on CIFAR-10 and test only at $\epsilon = 1\%$; TABOR (Guo et al., 2019) uses a 6x6 square as the trigger for GTSRB for images that are 32x32 (they additionally test on ImageNet but do not report good results until the trigger is over 25% of each dimension); and STRIP (Gao et al., 2019b) uses an 8x8 box on CIFAR-10. Our hypothesis is that the combination of a smaller trigger and more complex classes breaks defenses that demonstrate good performance in simpler contexts. Table 1 reports results using a single pixel trigger, which is often placed at the border of the image, within the region cropped by the standard random cropping data augmentation during training.

For comparison, we implemented the AC defense, which is included in the Adversarial Robustness Toolbox (Nicolae et al., 2019), an open-source collection of tools for security in machine learning. The authors report that AC achieves nearly perfect performance on two popular settings, namely, MNIST and traffic signs. We also implemented ITLM, which was tested on CIFAR-10 at $\epsilon = 5\%$ using larger L- and X-shaped triggers. Our results in Table 1 indicate that both AC and ITLM fail to completely defend against the poison in every setting, with the best targeted misclassification rate achieved by AC at 39.2% (compared to nearly 0% in every case with our defense).

To the best of our knowledge, SS is the only other defense in the literature which is evaluated using the same dataset (CIFAR-10) and class of triggers. The defense uses the eigenvectors of the feature matrix to separate clean and poisoned data. However, the authors do not appear to use triggers that can be cropped out during training by data augmentation. They also limit evaluation to "successfully" attacked networks, which they define as over 90% targeted misclassification rate of the undefended network (Column A, No defense). While our experiments suggest that SS is the strongest baseline after ours, successfully defending against the poison in 6 of the 12 scenarios considered in Table 1, its performance is poor particularly at lower $\epsilon$. Our results suggest that SS may fail to defend against harder to learn triggers requiring more complex feature representations.

## 7 CONCLUSION

Backdoor data poisoning attacks on deep neural networks are an emerging class of threats in the growing landscape of deployed machine learning applications. Though defenses exist, our experiments suggest that they only work against narrowly defined adversaries and fail dramatically when evaluated using more subtle threat models.

We introduce a new approach to defending against backdoor attacks based on an analysis of a novel self-expansion property in the training data. For a poisoned dataset satisfying mild compatibility properties, we show that an ensemble of weak learners fit to self-expanding sets successfully removes the poisoned data. Empirically, our method is resilient to a strong version of the dirty label backdoor attack introduced by Gu et al. (2017), which successfully evades all the baseline defenses. We believe our analysis and techniques present a valuable addition to the toolbox for secure deep learning.

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

## A    DEFERRED PROOFS

**Lemma A.1.** *Let $S_N$ and $S_P$ be two sets satisfying Property 4.4, and let $S$ and $T$ be drawn from $S_N$ and $S_P$, respectively. Then for all $0 < \gamma \leq \alpha$, $S$ and $T$ are mutually strictly $\gamma$-incompatible.*

*Proof.* We will show that $T$ is $\gamma$-incompatible with $S$ for all $\gamma \leq \alpha$; mutual incompatibility follows by a symmetric argument. First, $T$ is $\alpha$-incompatible with $S$ by consequence of Property 4.4. Fix $\gamma < \alpha$. Our main approach will be to subsample $S$ twice: first, we sample $S$ at a rate of $\gamma/\alpha$ to create $S'$ such that $|S'| \approx \gamma/\alpha|S|$. Then $S'$ is $\alpha$-incompatible with $T$, so $\epsilon(S'|T, \alpha) \geq \epsilon(S', \alpha)$. However we need take some care to ensure $\gamma/\alpha|S|$ is an integer.

Define

$$\alpha' := \frac{\gamma|S|}{\lfloor \gamma|S|/\alpha \rfloor} \tag{15}$$

Then $\alpha' \geq \alpha$ and $\gamma/\alpha'|S| = \lfloor \gamma|S|/\alpha \rfloor$. Thus, we will subsample $S$ at a rate of $\gamma/\alpha'$ to create $S'$; by Assumption 4.3, $T$ is $\alpha'$-incompatible with $S'$. Note that the resulting training sets produced by this double subsampling procedure have size $\gamma|S|$ as desired.

Next, we will show that there exists some constant $c$ such that

$$\epsilon(S|T, \gamma) = c(\gamma/\alpha'|S|)^{-1}\mathbb{E}[\epsilon(S'|T, \alpha')] \tag{16}$$

$$\epsilon(S, \gamma) = c(\gamma/\alpha'|S|)^{-1}\mathbb{E}[\epsilon(S', \alpha')] \tag{17}$$

where the expectations are taken over the subset $S' \subset S$, $|S'| = \gamma/\alpha'|S|$. Then since $T$ is strictly $\alpha'$-incompatible with all such $S'$ by Property 4.4,

$$\epsilon(S|T, \gamma) = c|S'|^{-1}\mathbb{E}[\epsilon(S'|T, \alpha')] \tag{18}$$

$$> c|S'|^{-1}\mathbb{E}[\epsilon(S', \alpha')] \tag{19}$$

$$= \epsilon(S, \gamma) \tag{20}$$

which is what we wanted to prove.

Fix a training set $S''$, $|S''| = \gamma|S|$, and let $T$ be arbitrary. By Assumption 4.3 we have that $R_{emp}(\mathcal{A}(S'' \cup T); S'') = 0$ for all $T'$. Then conditioning on the training set $S''$, we have that

$$\mathbb{E}[\epsilon(S'|T, \alpha')|S''] = \mathbb{E}[R_{emp}(\mathcal{A}(S'' \cup T); S')|S''] \tag{21}$$

$$= \mathbb{E}[R_{emp}(\mathcal{A}(S'' \cup T); S' - S'')|S''] \tag{22}$$

where the expectation is over $S'$ such that $S'' \subset S'$. Now $S' - S''$ is a random variable consisting of $|S'| - |S''|$ independent draws from $S$ with replacement; thus Equation 22 is just equal to $|S'| - |S''|$ times the empirical risk of a random element in $S$. On the other hand, $R_{emp}(\mathcal{A}(S'' \cup T); S)$ is the empirical risk over all elements in $S$, or equivalently, $|S|$ times the empirical risk of a random element.

Since $T$ was arbitrary, summing over all possible training sets $S''$ yields the desired identities with

$$c = \frac{|S|}{|S'| - |S''|}. \tag{23}$$

$\square$

*Proof of Theorem 4.5.* We first prove a slightly more general result. Assume by way of contradiction that there exists a partition of $S^*$ into two nonempty sets $P$ and $Q$ that are mutually strictly incompatible.

Let $\epsilon^*$ be the expansion error of $S^*$. Recall that this means

$$|S^*|\epsilon^* = |S^*|\epsilon(S^*; \alpha) = \mathbb{E}[R_{emp}(\mathcal{A}(S'); S^*)] \tag{24}$$

where the expectation is taken over samples $S'$ of size $\alpha^{-1}|S^*|$ drawn from $S^*$ with replacement. Since the empirical risk is a linear function of $S^*$, we can decompose this last term as

$$\mathbb{E}[R_{emp}(\mathcal{A}(S'); S^*)] = \mathbb{E}[R_{emp}(\mathcal{A}(S'); P)] + \mathbb{E}[R_{emp}(\mathcal{A}(S'); Q)] \tag{25}$$

Given a set of training samples $S'$, we will denote the elements drawn from $P$ and $Q$ as $P' = S' \cap P$ and $Q' = S' \cap Q$, respectively. Now consider the term

$$\mathbb{E}[R_{emp}(\mathcal{A}(S'); P)] = \mathbb{E}[R_{emp}(\mathcal{A}(P' \cup Q'); P)] \tag{26}$$

If we fix $Q'$, then $P'$ is drawn uniformly at random from $P$ with replacement, where $|P'| = \alpha|S^*| - |Q'|$. Define

$$\alpha_{Q'} := \frac{\alpha|S^*| - |Q'|}{|P|} \tag{27}$$

which is the subsampling rate of $P'$ given $Q'$. Note that if $\alpha' \geq \alpha$, then $Q'$ is $\alpha'$-incompatible with $P'$ by Assumption 4.3, and otherwise $\alpha' < \alpha$ and $Q'$ is strictly $\alpha'$-incompatible with $P'$ by Lemma A.1. Thus,

$$\mathbb{E}[R_{emp}(\mathcal{A}(P' \cup Q'); P)] = \sum_{Q'} \Pr[Q']\mathbb{E}[R_{emp}(\mathcal{A}(P' \cup Q'); P)|Q'] \tag{28}$$

$$= |P| \sum_{Q'} \Pr[Q']\epsilon(P|Q'; \alpha_{Q'}) \tag{29}$$

$$> |P| \sum_{Q'} \Pr[Q']\epsilon(P; \alpha_{Q'}) \tag{30}$$

On the other hand, $\mathbb{E}[\alpha_{Q'}] = \alpha$, and since $\epsilon(P; \alpha)$ is convex in $\alpha$ by Assumption 4.3, we can apply Jensen's inequality to conclude

$$\sum_{Q'} Pr[Q']\epsilon(P; \alpha_{Q'}) \geq \epsilon(P; \alpha), \tag{31}$$

i.e.,

$$\mathbb{E}[R_{emp}(\mathcal{A}(P' \cup Q'); P)] > |P|\epsilon(P; \alpha). \tag{32}$$

Similarly,

$$\mathbb{E}[R_{emp}(\mathcal{A}(P' \cup Q'); Q)] > |Q|\epsilon(Q; \alpha). \tag{33}$$

Combining these two results yields

$$|S^*|\epsilon^* > |P|\epsilon(P; \alpha) + |Q|\epsilon(Q; \alpha). \tag{34}$$

Since $|P| + |Q| = |S^*|$, we have that at least one of $\epsilon(P; \alpha)$ or $\epsilon(Q; \alpha)$ must be less than $\epsilon^*$, which contradicts the optimality of $S^*$. Thus one of $P$ or $Q$ must be empty.

Finally, we note that by Property 4.4, the partition $P = S^* \cap S_P$ and $Q = S^* \cap S_N$ gives an incompatible partition, which yields the result. $\square$

*Proof of Theorem 4.8.* We begin with the simple observation that if Properties 4.4 and 4.6 hold for $S$, they also hold for $S \setminus S'$ for any set $S'$. Thus we are able to apply Theorem 4.5 at each step and so in fact $S_1, ..., S_N$ are all homogeneous. Additionally, since $p > 1/2$ and a component's vote is weighted by its size, a sufficient condition for success is when all the primary components vote correctly.

We start with the case when $\mathcal{A}$ is deterministic. Let $S_i$ and $S_j$ be a primary component and noise component, respectively. By strict incompatibility, we have that $R_{emp}(\mathcal{A}(S_i); S_j) > R_{emp}(\mathcal{A}(\emptyset); S_j) = |S_j|/2$. Thus $S_i$ votes 0 on $S_j$. Conversely, if $S_j$ is a primary component, then $R_{emp}(\mathcal{A}(S_i); S_j) < R_{emp}(\mathcal{A}(\emptyset); S_j) = |S_j|/2$, so $S_i$ votes 1 on $S_j$. Putting these together and using the fact that $p > 1/2$, we find that the noise components have weighted vote strictly less than $|S|/2$, while the primary components have weighted vote strictly greater than $|S|/2$, as required.

For the case when $\mathcal{A}$ is stochastic, we apply standard concentration bounds to our estimates of the empirical risk of each component (over the randomness in $\mathcal{A}$). Let $S_i$ and $S_j$ be a primary and noise component, respectively. Again, strict incompatibility gives $R_{emp}(\mathcal{A}(S_i); S_j) \geq R_{emp}(\mathcal{A}(\emptyset); S_j) + \delta|S_j| = (1/2 + \delta)|S_j|$. Define the sample average empirical risk

$$V_B := \frac{1}{B|S_j|} \sum_{b=1}^{B} R_{emp}(\mathcal{A}_b(S_i); S_j) \tag{35}$$

computed from $B$ samples over the randomness in $\mathcal{A}$. Then $\mathbb{E}[V_B] \geq (1/2 + \delta)$ and so by Hoeffding's inequality

$$\Pr[|V_B < 1/2] \leq \Pr[|V_B - (1/2 + \delta)| > \delta] \tag{36}$$

$$< 2\exp(-2\delta^2 B) \tag{37}$$

Thus with probability at least $1 - 2\exp(-2\delta^2 B)$, primary component $S_i$ votes 0 when $S_j$ is a noise component. By the same argument and using strict compatibility, the bound also holds for $S_i$ voting 1 when $S_i$ and $S_j$ are both from primary components. Putting this together, we recall that a sufficient condition for success of the algorithm occurs when all the primary components vote correctly on all components (both primary and noise), which happens with probability at least

$$[1 - 2\exp(-2\delta^2 B)]^{MN} \tag{38}$$

as claimed. $\qquad\square$

*Proof of Proposition 5.1.* The statement is more or less a direct consequence of the alternating convex minimization strategy. Recall first that since $\alpha = 1$ and $\eta = 0$, Lines 4 and 6 in Algorithm 2 have no effect.

We prove the statement in two steps. First we claim that

$$F(\theta_{t+1}, v_t; \beta_t) \leq F(\theta_t, v_t; \beta_t) \tag{39}$$

Note that $v_t$ in the optimization objective $F$ plays the role of $S_t$ in Algorithm 2. The inequality follows from the optimization on Line 5 in Algorithm 2, which sets $\theta_{t+1}$ to the empirical risk minimizer of the set $v_t$.

Next, we claim that

$$F(\theta_{t+1}, v_{t+1}; \beta_{t+1}) \leq F(\theta_{t+1}, v_t; \beta_t) \tag{40}$$

Since $0 \leq L(\cdot, \cdot) < c$, the optimal size of the set $S_t$ is $|v_t| = \beta_t |S|$. Since $\beta_t$ is decreasing, we have that $|v_{t+1}| \leq |v_t|$. Thus the number of elements in the trimmed empirical loss is non-increasing (Line 7).

Combining the two inequalities shows that the objective function is decreasing in $t$. Since $F(\theta_t, v_t; \beta_t)$ is a decreasing sequence bounded from below by zero, the monotone convergence theorem gives the second result.

$\qquad\square$

## B EXPERIMENTAL DETAILS

### B.1 DEFENSE SET UP AND HYPERPARAMETERS

**ISPL + Boosting (this work).** For our defense, we use the same set of hyperparameters across all experiments. We run 8 rounds of ISPL, each of which returns a component consisting of roughly 12% of the total samples. Let $p$ be the target percentage of samples over the remaining samples (i.e., $p \approx 1/(8 - i + 1)$ in the $i^{th}$ iteration). Then the number of iterations $N$ is set to $2 + \min(3, 1/p)$. $\beta$ starts at $3 * p$ in the first iteration, then drops linearly to its final value of $p$ over the next 2 iterations. When trimming the training set, we also additionally include the top $p/8$ samples per class to prevent the network from collapsing to a trivial solution. For the learning procedure $\mathcal{A}$, we use standard SGD, trained for 4 epochs per iteration, with a warm-up in the first iteration of 8 epochs. The expansion factor $\alpha$ is set to $1/4$, and the momentum factor $\eta$ is set to $0.9$.

We run ISPL 3 times to generate 24 weak learners. Each weak learner is trained for 40 epochs on its respective subset. For the boosting framework, each component votes on a per-sample basis. The sample is preserved if the modal vote equals the given label, with ties broken randomly.

We also include a final self-training step by training a fresh model for 100 epochs on the recovered samples. The main idea is that a model fit to the full "clean" training data can be used to test the excluded training data, thereby recovering additional consistent data which may have been originally excluded because the weak learners were fit to a small subset of data for fewer epochs. However, it may take several repetitions of training a model from scratch before this self-training process no longer identifies new samples to recover. Therefore, we use a simple self-paced learning algorithm to dynamically adjust the samples during training to limit the self-training to a single iteration. More explicitly, we start with the "clean" samples as returned by the boosting framework. Every 5 epochs, we update the training set to be the samples whose labels agree with the model's current predictions. Due to the relative frequency with which we resample the training set, we smooth the predictions by a momentum factor of $0.8$ so that the training process is less noisy. The samples used for training in the last epoch are returned as the defended dataset. In our experiments, this process decreases the false positive rate (and thus increases the clean accuracy) but does not materially affect the false negative rate (nor the targeted misclassification rate).

**Spectral Signatures.** We use the official implementation of the Spectral Signatures defense (Tran et al., 2018) by the authors, available on Github, except that we replace the training procedure with PyTorch (instead of Tensorflow 1.x as in the authors' original implementation). The authors suggest removing 1.5 times the maximum expected amount of poison from each class for the defense. We remove 20% of each class for $\epsilon = 5, 10\%$ and 30% of each class for $\epsilon = 20\%$. In selecting the layer for the activations, for the ResNet32 architecture, we use the input to the third block of the third layer (which matches the authors' implementation), and for the PreActResNet18 architecture, we use the input to the first block of the four layer (which was found empirically to remove the most poison on the first set of scenarios). We note that the authors indicate the defense should be fairly successful at any of the later layers of the network.

**Iterative Trimmed Loss Minimization.** The Iterative Trimmed Loss Minimization defense (Shen and Sanghavi, 2019) consists of an iterative procedure. Given a setting $0 < \alpha \le 1$, one first trains a model for a number of epochs. Then the $\alpha$ fraction of samples with the lowest loss are retained for the next iteration. This process is repeated several times, with a fresh model beginning each iteration. The defended dataset is the $\alpha$ fraction of samples with the lowest loss after the last iteration. For the backdoor data poisoning experiments on CIFAR-10, the authors use 80 epochs for the first round of training, then 40 epochs thereafter; they also set $\alpha = 98\%$ for $\epsilon = 5\%$, and do not test at other values of $\epsilon$. We use the same settings, and scale $\alpha$ linearly with $\epsilon$, i.e., $\alpha = 96\%$ for $\epsilon = 10\%$ and $\alpha = 92\%$ for $\epsilon = 20\%$.

**Activation Clustering.** The Activation Clustering defense (Chen et al., 2018) has an actively maintained official implementation in the Adversarial Robustness Toolbox (ART) (Nicolae et al., 2019), an open-source collection of tools for security in machine learning. We use the official implementation with the default parameters values in ART v1.6.2, the most current version at the

time of writing. In selecting the layer for the activations, we used the same layer as for Spectral Signatures.

**Models.** The PreActResNet18 He et al. (2016b) model is optimized using vanilla SGD with learning rate 0.02, momentum 0.9, and weight decay 5e-4. For the final dataset, we train for 200 epochs and drop the learning rate by 10 at epochs 100, 150, and 180. Using these parameters, we achieve 94.7% accuracy on CIFAR-10 when trained and tested with clean data.

The ResNet32 He et al. (2016a) model is optimized using vanilla SGD with learning rate 0.1, momentum 0.9, and weight decay 1e-4. For the final dataset, we train for 200 epochs and drop the learning rate by 10 at epochs 100 and 150. Using these parameters, we achieve 91.8% accuracy on CIFAR-10 when trained and tested with clean data.

## B.2    BACKDOOR POISON DATASET CONSTRUCTION

Each scenario has a single source and target class. We use the same (source, target) pairs as in Tran et al. (2018): (airplane, bird), (automobile, cat), (bird, dog), (cat, dog), (cat, horse), (horse, deer), (ship, frog), (truck, bird).

To generate a perturbation, we choose a shape (L-shape, X-shape, or pixel) uniformly at random. The (X,Y) coordinates of the perturbation are randomly selected to guarantee that the entire shape is visible before data augmentation (e.g., the pixel-based perturbation can be placed anywhere within the 32x32 image, but the X-shape is larger and so must be centered in a 30x30 region, one pixel away from the border). The color of the perturbation is also selected uniformly at random, with each of the (R,G,B) coordinates ranging from 0 to 255. Finally, we randomly select an $\epsilon = 5, 10, 20\%$ percentage of the source class, apply the perturbation by replacing the pixels in the corresponding locations with the selected shape and color, then relabel the poisoned images as the target class.

Table 2 displays the generated triggers used in our experiments with examples of poisoned images. Within the row for each (source, target) pair, the first subrow gives the parameters for poison 1, the second subrow gives the parameters for poison 2, and the third subrow gives the parameters for poison 3. We also provide an example of the corresponding clean image for poison 1 in column clean 1. Note that the results presented in Table 1 of the main paper use the first scenario of each (source, target) pair (poison 1).

Table 2: CIFAR-10 dirty label backdoor scenarios.

| source | target | color | position | method | clean 1 | poison 1 | poison 2 | poison 3 |
|--------|--------|-------|----------|--------|---------|----------|----------|----------|
| 0 / Plane | 2 / Bird | (103, 87, 79) | (24, 3) | pixel | | | | |
| | | (92, 1, 189) | (27, 30) | pixel | | | | |
| | | (47, 2, 21) | (21, 8) | pixel | | | | |
| 1 / Car | 3 / Cat | (180, 98, 53) | (10, 30) | pixel | | | | |
| | | (40, 105, 92) | (25, 13) | pixel | | | | |
| | | (145, 70, 200) | (9, 29) | pixel | | | | |
| 2 / Bird | 5 / Dog | (93, 86, 130) | (27, 9) | pixel | | | | |
| | | (156, 158, 244) | (18, 30) | pixel | | | | |
| | | (74, 162, 26) | (11, 9) | pixel | | | | |
| 3 / Cat | 5 / Dog | (34, 241, 240) | (2, 14) | L | | | | |
| | | (239, 42, 58) | (28, 13) | pixel | | | | |
| | | (39, 221, 162) | (7, 23) | X | | | | |
| 3 / Cat | 7 / Horse | (61, 14, 183) | (16, 2) | X | | | | |
| | | (180, 50, 21) | (11, 0) | pixel | | | | |
| | | (4, 221, 78) | (24, 22) | L | | | | |
| 7 / Horse | 4 / Deer | (107, 60, 58) | (15, 4) | pixel | | | | |
| | | (242, 30, 233) | (4, 21) | X | | | | |
| | | (76, 14, 15) | (11, 19) | L | | | | |
| 8 / Ship | 6 / Frog | (141, 245, 211) | (4, 1) | pixel | | | | |
| | | (213, 221, 138) | (19, 29) | X | | | | |
| | | (121, 158, 6) | (3, 13) | pixel | | | | |
| 9 / Truck | 2 / Bird | (187, 67, 135) | (4, 27) | pixel | | | | |
| | | (69, 204, 11) | (14, 29) | L | | | | |
| | | (239, 186, 219) | (1, 29) | X | | | | |

Table 3: Performance on CIFAR-10, dirty label backdoor scenario, using the PreActResNet18 architecture. The S / T column lists the CIFAR-10 source and target classes. $\epsilon$ refers to the percentage of the source class which is poisoned. For the remainder of the columns, the top level column headers give the defense type: L (clean), ND (no defense), SS (spectral signatures), AC (activation clustering), ITLM (iterative trimmed loss minimization, and TW (this work); the second level column headers give the metric type: M (misclassification rate), C (clean accuracy, higher is better), A (targeted misclassification rate, lower is better), FP (false positives, lower is better), FN (false negatives, lower is better). Please refer to the text for a more detailed explanation of the table.

| S / T | $\epsilon$ | L | ND | | SS | | | | AC | | | | ITLM | | | | TW | | | |
|---|---|---|---|---|---|---|---|---|---|---|---|---|---|---|---|---|---|---|---|---|
| | | M | C | A | C | A | FP | FN | C | A | FP | FN | C | A | FP | FN | C | A | FP | FN |
| | 5 | 1.3 | 94.5 | 91.3 | 94.5 | 79.9 | 7381 | 130 | 91.9 | 58.3 | 18926 | 155 | 94.6 | 84.9 | 994 | 244 | 92.8 | 0.0 | 3640 | 23 |
| 0 / 2 | 10 | 1.3 | 94.1 | 90.6 | 94.5 | 66.0 | 7383 | 383 | 92.3 | 85.9 | 19790 | 320 | 94.2 | 92.8 | 1961 | 461 | 92.8 | 0.0 | 3479 | 25 |
| | 20 | 1.1 | 94.6 | 80.2 | 94.2 | 64.2 | 14335 | 335 | 92.4 | 73.9 | 19127 | 601 | 93.8 | 93.8 | 3897 | 897 | 91.6 | 22.9 | 3711 | 237 |
| | 5 | 0.0 | 94.4 | 92.9 | 94.7 | 5.5 | 7319 | 68 | 90.9 | 19.5 | 19791 | 155 | 94.8 | 96.5 | 986 | 236 | 92.8 | 0.0 | 3434 | 2 |
| 1 / 3 | 10 | 0.0 | 94.6 | 98.4 | 94.5 | 0.0 | 7035 | 35 | 92.5 | 68.7 | 19033 | 333 | 94.6 | 98.0 | 1981 | 481 | 93.0 | 0.0 | 3348 | 3 |
| | 20 | 0.0 | 94.4 | 99.6 | 94.6 | 0.0 | 14007 | 7 | 92.2 | 91.8 | 18317 | 622 | 94.4 | 99.4 | 3914 | 914 | 93.0 | 0.0 | 3253 | 2 |
| | 5 | 1.1 | 94.4 | 80.4 | 94.6 | 76.4 | 7494 | 243 | 92.4 | 53.9 | 19937 | 172 | 94.7 | 79.4 | 985 | 235 | 92.6 | 0.2 | 3704 | 5 |
| 2 / 5 | 10 | 0.9 | 94.4 | 97.2 | 94.4 | 0.1 | 7053 | 53 | 92.9 | 81.8 | 18657 | 313 | 94.7 | 92.4 | 990 | 490 | 92.9 | 0.2 | 3200 | 16 |
| | 20 | 1.2 | 94.4 | 94.0 | 94.5 | 87.7 | 14263 | 263 | 92.6 | 89.4 | 20531 | 406 | 94.6 | 95.4 | 966 | 966 | 92.8 | 0.3 | 3540 | 38 |
| | 5 | 7.6 | 94.7 | 91.0 | 94.7 | 88.5 | 7281 | 30 | 92.6 | 90.8 | 19172 | 172 | 94.5 | 91.4 | 993 | 243 | 92.8 | 81.1 | 3693 | 167 |
| 3 / 5 | 10 | 5.9 | 94.8 | 94.0 | 94.4 | 8.6 | 7014 | 14 | 92.3 | 91.6 | 20293 | 296 | 94.6 | 92.3 | 988 | 488 | 92.6 | 90.2 | 3355 | 496 |
| | 20 | 7.6 | 94.7 | 90.8 | 94.3 | 90.6 | 14092 | 92 | 92.6 | 92.6 | 18236 | 652 | 94.7 | 91.7 | 974 | 974 | 92.6 | 87.4 | 3210 | 995 |
| | 5 | 0.6 | 94.3 | 33.8 | 94.6 | 98.3 | 7500 | 249 | 91.9 | 97.2 | 21289 | 246 | 94.7 | 98.4 | 995 | 245 | 92.7 | 0.0 | 3692 | 5 |
| 3 / 7 | 10 | 0.7 | 94.4 | 98.5 | 94.6 | 96.6 | 7135 | 135 | 92.2 | 98.2 | 20630 | 484 | 94.5 | 98.6 | 1979 | 479 | 92.7 | 2.4 | 3427 | 22 |
| | 20 | 0.7 | 94.4 | 98.5 | 93.7 | 78.5 | 14675 | 675 | 92.3 | 98.2 | 20470 | 987 | 94.4 | 98.9 | 980 | 980 | 92.9 | 10.5 | 3259 | 43 |
| | 5 | 1.5 | 94.7 | 92.0 | 94.6 | 0.6 | 7280 | 29 | 92.3 | 39.2 | 19353 | 150 | 94.8 | 91.6 | 992 | 242 | 92.8 | 0.5 | 3258 | 30 |
| 7 / 4 | 10 | 1.5 | 94.6 | 94.7 | 94.4 | 0.0 | 7023 | 23 | 92.0 | 47.9 | 19631 | 285 | 94.4 | 94.5 | 1979 | 479 | 93.0 | 0.3 | 3627 | 293 |
| | 20 | 1.5 | 94.6 | 96.5 | 94.3 | 0.1 | 14000 | 0 | 92.1 | 78.6 | 21292 | 22 | 94.4 | 96.5 | 3910 | 910 | 92.8 | 84.5 | 3203 | 867 |
| | 5 | 0.2 | 94.7 | 97.0 | 94.8 | 80.5 | 7470 | 219 | 92.8 | 73.3 | 19293 | 152 | 94.5 | 98.3 | 993 | 243 | 92.8 | 0.0 | 3494 | 2 |
| 8 / 6 | 10 | 0.2 | 94.4 | 99.5 | 94.7 | 0.0 | 7008 | 8 | 92.6 | 97.6 | 19544 | 288 | 94.4 | 99.4 | 1981 | 481 | 92.9 | 0.0 | 3571 | 1 |
| | 20 | 0.2 | 94.7 | 99.5 | 94.4 | 0.0 | 14000 | 0 | 92.5 | 96.4 | 18946 | 597 | 94.2 | 99.5 | 3908 | 908 | 92.6 | 0.2 | 3492 | 8 |
| | 5 | 0.1 | 95.0 | 92.0 | 94.4 | 93.3 | 7501 | 250 | 91.7 | 85.0 | 23573 | 151 | 94.6 | 97.3 | 988 | 238 | 93.0 | 0.0 | 3291 | 2 |
| 9 / 2 | 10 | 0.1 | 94.5 | 93.9 | 94.3 | 93.1 | 7500 | 500 | 92.1 | 95.1 | 22896 | 251 | 94.4 | 98.6 | 1970 | 471 | 93.1 | 0.0 | 3133 | 1 |
| | 20 | 0.1 | 94.4 | 96.1 | 94.7 | 0.0 | 14010 | 10 | 93.1 | 98.9 | 18651 | 575 | 94.1 | 99.0 | 3906 | 906 | 93.1 | 0.0 | 3223 | 2 |

## C  ADDITIONAL EXPERIMENTAL RESULTS

Tables 3 and 4 summarizes our main results for all the (source, target) pairs using two standard architectures for image classification: a PreActResNet18 He et al. (2016b) network and a ResNet32 He et al. (2016a) network, respectively.

For each (source, target) pair, we generated three scenarios. For each (source, target) pair and setting of epsilon, we report results for the scenario in which the defense's targeted misclassification rate (column A) was the median of all three scenarios. For the clean and no defense columns, we report results for the same scenario as TW (This Work).

The set of defenses consists of

1. (L) Clean, training on the entire clean training set. We report only the misclassification rate (M), which is the number of poisoned samples from the test set of the source class that are misclassified as the target class.

2. (ND) No Defense, training on entire poisoned training set. We report only the clean accuracy (C) and targeted misclassification rate (A) in this case.

3. (SS) Spectral Signatures (Tran et al., 2018)

4. (AC) Activation Clustering (Chen et al., 2018)

Table 4: Performance on CIFAR-10, dirty label backdoor scenario, using the ResNet32 architecture. The S / T column lists the CIFAR-10 source and target classes. $\epsilon$ refers to the percentage of the source class which is poisoned. For the remainder of the columns, the top level column headers give the defense type: L (clean), ND (no defense), SS (spectral signatures), AC (activation clustering), ITLM (iterative trimmed loss minimization, and TW (this work); the second level column headers give the metric type: M (misclassification accuracy), C (clean accuracy, higher is better), A (targeted misclassification rate, lower is better), FP (false positives, lower is better), FN (false negatives, lower is better). Please refer to the text for a more detailed explanation of the table.

| S / T | $\epsilon$ | L M | ND C | ND A | SS C | SS A | SS FP | SS FN | AC C | AC A | AC FP | AC FN | ITLM C | ITLM A | ITLM FP | ITLM FN | TW C | TW A | TW FP | TW FN |
|---|---|---|---|---|---|---|---|---|---|---|---|---|---|---|---|---|---|---|---|---|
| 0 / 2 | 5 | 1.1 | 92.6 | 85.0 | 91.8 | 57.3 | 7499 | 248 | 88.8 | 0.6 | 24211 | 133 | 91.8 | 83.3 | 988 | 238 | 89.8 | 0.0 | 5752 | 31 |
| | 10 | 1.1 | 92.4 | 92.6 | 91.7 | 91.5 | 7422 | 422 | 88.8 | 78.6 | 24031 | 250 | 91.7 | 92.3 | 1969 | 469 | 89.5 | 0.0 | 5445 | 33 |
| | 20 | 1.5 | 92.5 | 94.9 | 90.4 | 64.2 | 14590 | 590 | 88.6 | 62.7 | 23686 | 455 | 91.9 | 94.2 | 3890 | 890 | 88.0 | 0.0 | 6992 | 86 |
| 1 / 3 | 5 | 0.1 | 92.7 | 98.5 | 91.7 | 5.7 | 7479 | 227 | 88.8 | 2.4 | 23903 | 125 | 91.9 | 12.7 | 988 | 238 | 89.8 | 0.0 | 5776 | 1 |
| | 10 | 0.1 | 91.6 | 35.0 | 91.1 | 3.3 | 7436 | 436 | 88.3 | 24.9 | 23891 | 254 | 92.0 | 69.2 | 988 | 488 | 89.8 | 0.0 | 5736 | 3 |
| | 20 | 0.0 | 92.1 | 98.1 | 91.5 | 0.0 | 14004 | 4 | 89.1 | 0.4 | 21145 | 70 | 91.9 | 97.3 | 3900 | 900 | 89.4 | 0.0 | 5833 | 1 |
| 2 / 5 | 5 | 1.7 | 92.2 | 62.7 | 91.2 | 43.2 | 7493 | 242 | 88.0 | 1.4 | 24001 | 128 | 92.2 | 75.7 | 994 | 244 | 88.8 | 0.1 | 6358 | 6 |
| | 10 | 1.6 | 92.5 | 92.4 | 91.9 | 89.8 | 7476 | 476 | 89.3 | 81.5 | 23827 | 275 | 91.6 | 93.2 | 1975 | 475 | 89.2 | 0.0 | 5985 | 26 |
| | 20 | 1.6 | 91.7 | 95.8 | 89.9 | 3.5 | 14349 | 349 | 88.5 | 87.2 | 21086 | 711 | 91.9 | 95.1 | 3905 | 905 | 89.8 | 0.2 | 5684 | 40 |
| 3 / 5 | 5 | 6.3 | 91.3 | 88.9 | 91.7 | 87.9 | 7466 | 215 | 88.9 | 80.0 | 23817 | 131 | 92.2 | 90.8 | 996 | 246 | 89.2 | 27.5 | 5974 | 59 |
| | 10 | 6.2 | 91.8 | 90.8 | 91.6 | 86.4 | 7348 | 348 | 88.7 | 73.1 | 21299 | 136 | 92.2 | 89.5 | 990 | 490 | 89.1 | 82.1 | 5681 | 344 |
| | 20 | 6.3 | 92.3 | 90.8 | 90.5 | 71.9 | 14090 | 90 | 89.3 | 78.5 | 20694 | 156 | 90.9 | 88.8 | 3918 | 918 | 89.8 | 86.7 | 5093 | 994 |
| 3 / 7 | 5 | 1.1 | 92.6 | 98.4 | 91.0 | 97.1 | 7452 | 201 | 86.2 | 72.8 | 23823 | 189 | 92.1 | 97.1 | 994 | 244 | 89.4 | 0.2 | 5498 | 13 |
| | 10 | 1.1 | 92.4 | 98.6 | 91.8 | 96.7 | 7315 | 315 | 88.6 | 96.2 | 23648 | 482 | 92.3 | 98.0 | 1981 | 481 | 89.8 | 0.0 | 5126 | 16 |
| | 20 | 1.1 | 92.9 | 98.6 | 90.9 | 97.0 | 14167 | 167 | 89.1 | 95.4 | 20600 | 551 | 92.4 | 98.1 | 3924 | 924 | 88.6 | 0.2 | 6642 | 26 |
| 7 / 4 | 5 | 1.7 | 92.1 | 88.5 | 91.6 | 87.5 | 7486 | 235 | 89.3 | 72.2 | 23928 | 149 | 92.4 | 92.2 | 992 | 242 | 88.8 | 0.4 | 6643 | 27 |
| | 10 | 1.7 | 92.7 | 93.9 | 91.9 | 94.2 | 7371 | 371 | 88.3 | 59.2 | 23753 | 193 | 92.1 | 96.2 | 1973 | 473 | 88.9 | 0.8 | 6478 | 32 |
| | 20 | 1.7 | 92.6 | 96.9 | 91.1 | 94.1 | 14397 | 397 | 88.2 | 47.6 | 23737 | 423 | 91.7 | 95.8 | 3904 | 904 | 88.6 | 46.9 | 6322 | 209 |
| 8 / 6 | 5 | 0.2 | 92.7 | 98.0 | 91.3 | 97.8 | 7441 | 190 | 89.1 | 88.7 | 23658 | 164 | 92.5 | 96.9 | 988 | 238 | 89.6 | 0.0 | 5991 | 0 |
| | 10 | 0.2 | 92.1 | 97.7 | 91.8 | 97.2 | 7089 | 89 | 90.2 | 95.1 | 19585 | 280 | 92.3 | 98.7 | 973 | 473 | 89.3 | 0.0 | 6007 | 2 |
| | 20 | 0.2 | 92.8 | 98.8 | 91.3 | 96.0 | 14297 | 297 | 87.2 | 64.2 | 23347 | 646 | 92.3 | 98.6 | 975 | 975 | 89.4 | 0.0 | 5691 | 3 |
| 9 / 2 | 5 | 0.1 | 92.9 | 93.2 | 91.2 | 93.2 | 7478 | 225 | 88.7 | 1.2 | 23518 | 136 | 92.1 | 94.0 | 991 | 241 | 90.3 | 0.0 | 5444 | 2 |
| | 10 | 0.1 | 92.6 | 98.6 | 91.4 | 94.7 | 7497 | 497 | 88.5 | 91.5 | 24034 | 242 | 92.5 | 97.8 | 986 | 486 | 88.1 | 0.0 | 7220 | 2 |
| | 20 | 0.1 | 92.6 | 97.4 | 90.5 | 0.0 | 14011 | 11 | 90.6 | 97.7 | 18658 | 578 | 91.9 | 99.2 | 3881 | 811 | 89.6 | 0.0 | 5742 | 1 |

5. (ITLM) Iterative Trimmed Loss Minimization (Shen and Sanghavi, 2019)

6. (TW) This Work

For each defense, we report

1. (C) clean accuracy, which is the accuracy of the defended network on the entire clean test set (higher is better).

2. (A) targeted misclassification rate as defined in Equation 7, which is measured over the entire source class of the test set (lower is better).

3. (FP) false positives, which counts the number of clean samples excluded from the defended training set (lower is better).

4. (FN) false negatives, which counts the number of poisoned samples included in the defended training set (lower is better).

### C.1 DISCUSSION

Our approach consistently outperforms all other defenses by targeted misclassification rate (column A) across both architectures. If we define a "successful" run as achieving less than 1% targeted misclassification rate, then for the PreActResNet18 architecture, our defense succeeds in 17/24 scenarios, SS succeeds 9/24 scenarios, and both AC and ITLM do not succeed a single time; for the ResNet32 architecture, our defense succeeds in 20/24 scenarios, SS succeeds in 2/24 scenarios, AC succeeds in 1/24 scenarios, and ITLM again fails all 24 scenarios.

In general, our defense results in a 2-3% drop in clean accuracy for both architectures, when compared to a model trained and tested using clean data. AC achieves a clean accuracy which is on par with (or slightly below) ours. Surprisingly, this clean accuracy is despite AC having false positives (FP) of approximate 6x and 4x ours for the PreActResNet18 and ResNet32 models, respectively. Similarly, compared to our defense, SS has a slightly higher FP rate (which is roughly constant, as the defense always removes a fixed amount of data), but suffers a negligible drop in clean accuracy. We attribute this behavior to the existence of small, difficult to learn subpopulations (that may be removed by the weak learners as incompatible after training for only 40 epochs) but are responsible for the last 2-3% of performance. However, we note that our defense is designed to remove incompatible data, rather than poisoned data specifically, and therefore some such behavior is expected. Conversely, we hypothesize that SS and AC are removing "easy" data according to statistical properties of the activation patterns of a trained network, which may constitute redundant data in terms of the training distribution. ITLM achieves good clean accuracy and the lowest number of false positives (though its performance is negligible in terms of defending against poison).

The only scenario which consistently evades our defense is the (3 / Cat, 5 / Dog) scenario. This scenario is also the only one for which the poison misclassification rate of a clean network is noticeable large at around 6-8% (primary column L, secondary column M), which is consistent with the results in Tran et al. (2018). These results suggest that the scenario violates Property 4.4, i.e., the poison and clean distributions are not incompatible—training on a clean dataset yields non-negligible performance on poisoned cats when mislabeled as dogs. Because the poisoned data is compatible with the clean data, our theoretical analysis suggests that our defense will struggle to separate the clean and poisoned data, as is reflected in our results. Despite this, we note that the performance of our defense still exceeds that of the SS, AC, and ITLM defenses in several cases for this scenario.

Finally, to reconcile our results with the results presented in the Spectral Signatures paper, we note that the training code in official implementation of the SS defenses uses some non-standard methodologies, including a random crop with only 2x2 padding (instead of the 4x4 commonly used for CIFAR-10); no normalization of the input data according to the mean and standard deviation; and custom initialization of all the layers (such as using a normal distribution to initialize the convolutional layers, rather than the default Kaiming initialization (He et al., 2015) in PyTorch). The authors also only report results for cases where the network was "successfully poisoned", which they defined as "approximately 90% or higher accuracy on the poisoned set" (corresponding to primary column ND, secondary column A, in Tables 3 and 4). To verify our results, we ran the first scenario of the first (source, target) pair (i.e., the first row of Table 2) through the authors' own implementation and found that at $\epsilon = 5\%$

- an undefended network had a 71.9% poison misclassification rate;
- the defense left 205 false negatives (out of 250 poisoned images);
- trained on the defended dataset, the network had a 52.9% poison misclassification rate,

and at $\epsilon = 10\%$

- an undefended network had a 74.1% poison misclassification rate;
- the defense left 193 false negatives (out of 500 poisoned images);
- trained on the defended dataset, the network had a 23.3% poison misclassification rate.

These results are not within the scope of the results considered in the original paper (due to not being over 90% poisoned pre-defense). In contrast, in our experiments, the pre-defense poison misclassification rate is much higher, which we attribute to more modern training methodologies.

Table 5: Ablation studies on CIFAR-10, dirty label backdoor scenario, using the PreActResNet18 architecture, with various settings of $\alpha$ and $\beta$. The S / T column lists the CIFAR-10 source and target classes. $\epsilon$ refers to the percentage of the source class which is poisoned. The second level headings are C (clean accuracy, higher is better), A (targeted misclassification rate, lower is better). Please refer to the text for a more detailed explanation of the table.

|  | S / T | $\epsilon$ | $\beta = 1/16$ | | $\beta = 1/8$ | | $\beta = 1/4$ | | $\beta = 1$ | |
|---|---|---|---|---|---|---|---|---|---|---|
|  |  |  | C | A | C | A | C | A | C | A |
| $\alpha = 1/4$ | 3 / 5 | 5 | 92.7 | 79.8 | 92.8 | 81.1 | 92.9 | 71.8 | 93.0 | 55.7 |
|  |  | 10 | 92.6 | 87.3 | 92.6 | 90.2 | 93.1 | 89.9 | 93.5 | 91.9 |
|  |  | 20 | 92.3 | 76.1 | 92.6 | 87.4 | 92.8 | 88.4 | 93.3 | 92.3 |
|  | 7 / 4 | 5 | 92.9 | 2.4 | 92.8 | 0.5 | 93.3 | 0.2 | 93.3 | 48.2 |
|  |  | 10 | 92.6 | 9.3 | 92.9 | 0.3 | 93.2 | 27.6 | 93.2 | 76.7 |
|  |  | 20 | 92.7 | 83.5 | 92.8 | 84.5 | 92.8 | 84.7 | 93.2 | 43.0 |
|  | 8 / 6 | 5 | 92.8 | 0.0 | 93.1 | 0.0 | 93.3 | 0.0 | 93.4 | 0.0 |
|  |  | 10 | 92.7 | 0.0 | 93.0 | 0.0 | 93.3 | 0.0 | 93.5 | 0.1 |
|  |  | 20 | 92.8 | 0.1 | 92.6 | 0.2 | 93.1 | 0.0 | 93.1 | 95.5 |
|  | 9 / 2 | 5 | 92.9 | 0.0 | 93.0 | 0.0 | 93.3 | 0.0 | 93.2 | 0.0 |
|  |  | 10 | 92.8 | 0.0 | 92.9 | 0.0 | 92.9 | 0.0 | 92.8 | 0.0 |
|  |  | 20 | 92.7 | 0.0 | 93.1 | 0.0 | 93.0 | 0.0 | 93.3 | 0.0 |
| $\alpha = 1$ | 3 / 5 | 5 | 92.8 | 80.6 | 92.8 | 74.7 | 93.0 | 73.6 | 92.7 | 87.1 |
|  |  | 10 | 92.4 | 86.8 | 92.3 | 86.1 | 92.8 | 91.4 | 93.0 | 90.1 |
|  |  | 20 | 91.5 | 79.8 | 92.6 | 85.3 | 93.0 | 87.6 | 93.1 | 90.4 |
|  | 7 / 4 | 5 | 92.6 | 0.9 | 93.4 | 0.9 | 92.9 | 5.7 | 93.5 | 0.2 |
|  |  | 10 | 92.9 | 2.3 | 92.8 | 72.1 | 93.1 | 0.5 | 93.5 | 72.1 |
|  |  | 20 | 92.7 | 91.7 | 92.6 | 83.4 | 92.9 | 88.4 | 93.0 | 86.6 |
|  | 8 / 6 | 5 | 92.8 | 0.0 | 92.2 | 0.0 | 92.8 | 0.0 | 93.3 | 0.0 |
|  |  | 10 | 92.8 | 0.1 | 93.1 | 0.0 | 93.2 | 0.0 | 93.5 | 0.0 |
|  |  | 20 | 92.3 | 0.0 | 92.7 | 0.0 | 93.5 | 0.0 | 93.2 | 0.0 |
|  | 9 / 2 | 5 | 92.6 | 0.0 | 92.9 | 0.0 | 93.1 | 0.0 | 93.1 | 0.0 |
|  |  | 10 | 92.8 | 0.0 | 93.0 | 0.0 | 93.1 | 0.0 | 93.0 | 0.0 |
|  |  | 20 | 92.8 | 0.5 | 92.8 | 0.0 | 92.6 | 0.0 | 93.2 | 0.0 |

## C.2 ABLATION STUDIES

We conduct some additional ablation studies to better understand the effects of the two main hyper-parameters in Algorithm 2: the expansion factor $\alpha$ and the subset size $\beta$. Computationally, larger $\beta$ means fewer components and fewer outer iterations of ISPL (and is thus more efficient); in our main experiments, we use $\beta = 1/8$ and run ISPL 8 times in sequence to generate 8 components. Additionally, as discussed in Remark 2, smaller $\alpha$ is a more stringer requirement, since mixing distributions increases the expansion factor. Therefore we would expect that increasing $\alpha$ leads to worse identification of homogeneous components on average.

Tables 5 presents the full results of the ablation studies. Our main finding is that our method is quite robust to both the expansion factor $\alpha$ and subset size $\beta$. There is also a slight trend that smaller $\alpha$ and $\beta$ are better at identifying poison, with a small drop in clean accuracy.

