# OpenReview forum: "Defending Against Backdoor Attacks Using Ensembles of Weak Learners "
_ICLR.cc/2022/Conference — ICLR 2022 Submitted_

### Official Review · Reviewer_cvHA · 2021-10-31

**Correctness:** 2
**Technical Novelty And Significance:** 2
**Empirical Novelty And Significance:** 2
**Recommendation:** 3
**Confidence:** 4

**Details Of Ethics Concerns:**

No ethics concerns.

**Main Review:**

Strong points:
1. The idea of data partitioning, assessing then reassembling is generally new for adversarial defense.

2. The proposed method seems more effective than prior work the Iterative Trimmed Loss Minimization (ITLM) of Shen and Sanghavi, 2019.

3. Theoretical analyses are provided to justify the proposed approach.


Weak points:

1. The threat model adopted in this work needs more justifications. Backdoor attacks are no longer purely data poisoning anymore, there are quite a number of them are “model poisoning” attacks that follow a different setting. They follow different threat models in terms of who controls the training process. I understand this paper follows the more traditional data poisoning threat model, but it needs to be made very clear and contrasted to other backdoor defense threat models.


2. Where the authors place their defense in the current literature is not clear. It appears to me that the proposed method is a robust training method, not a detection or backdoor removal method. The defense scope should also be discussed explicitly. The proposed method can only defense dirty-label backdoors with simple trigger patterns.

3. Some of the assumptions are too strong, and their implications to backdoor attack/defense should also be explained. Property 4.4 (Incompatibility of primary and noise distributions), this assumption e most excludes most advanced backdoor attacks (trojan [1], dynamic input-aware [2], sample-wise [3], clean-label [4], feature-space attack, invisible attack, blending attack, reflection attack, and many others), and limits it to only simple attacks like BadNets. The authors can find many other examples in this GitHub repo: https://github.com/THUYimingLi/backdoor-learning-resources. Backdoor examples are *by design* compatible to clean data distribution, iconic characteristic of backdoor attacks to traditional data poisoning attacks. The tested BadNets is a classific dirty-label backdoor attack, which is very close to the traditional poisoning attacks but can no longer represent all backdoor attacks.

4. Theorem 4.5 (Sets minimizing expansion error are homogeneous) is based on the strong assumption of incompatibility. I doubt it will hold for backdoor attacks, not just because of the incompatibility assumption, but also in general without knowing the poisoning rate.  The authors can prove me wrong with 40% poisoning but keeping the same number of samples. How to calculate the number of remaining/removed samples anyway?

5. How does Algorithm 1 help is not well explained. The input has homogenous sets S_1-S_N, the return is their votes (based on their mean empirical error), is this to choose low-loss samples like in ITLM, Shen and Sanghavi, 2019. How does this differ from their strategy except using the ensemble (but in the full set case)? How the votes contribute to Algorithm 2?

6. In Proof of Theorem 4.5, $S^{‘}=P^{‘} \cup Q^{‘}$, why?

7. What is the overall training procedure? How time-consuming it is compared to standard training. The training details given in the appendix show that the proposed method is complex, ad hoc, and hard to tune for other datasets, attacks, or poisoning rates.

8. Confusing notations: $\alpha-$expansion/compatibility, $\beta$-compatibility, and the purposes of the samples determined by $\alpha$ and $\beta$ in Algorithm 2.

9. The biggest concern about the experiment, except for missing many different types of attacks and datasets, is the cherry-picking results in the main text, Table 1. The complete results in appendix Table 3-5 indicate many failure cases on some of the source-target classes, for example, 3/5, 7/4, while the results in Table 1 only show the successful ones. So, based on the more complete results, the proposed defense does work at least on some source-traget class pairs. I doubt it will work on high poisoning rate or other types of advanced attacks like input-aware, sample-wise, clean labels attacks.

10. Missing discussion of other defense methods: detection methods and backdoor removal methods. Can detection methods be directly applied to detect and remove the backdoor examples, if one can retrain the model on the purified data? Can backdoor trigger removal methods like Model be used or compared? What's wrong with the assumption of having a small set of clean samples, if the authors assume the defender has access to the full dataset and the dataset is less than 50% poisoned?

11. The authors kept arguing that many existing defense works are not tested on CIFAR-10, why the authors didn’t test other datasets since many of them did? What makes the CIFAR-10 dataset so special? Transferability of the developed defense to large datasets? CIFAR-10 is not a good backdoor dataset is simply because its resolution is low, which is not an ideal place to design and test different types of trigger patterns, and even they work on CIFAR-10, they may not transfer to real-world scenarios which are surely not low resolution.

---
[1] Trojaning attack on neural networks, Liu et al.

[2] Input-Aware Dynamic Backdoor Attack, NeurIPS, 2020.

[3] Invisible Backdoor Attack with Sample-Specific Triggers, ICCV 2021. (maybe concurrent work)

[4] Clean-Label Backdoor Attacks, Turner, Tsipras and Madry.

[5] Bridging Mode Connectivity in Loss Landscapes and Adversarial Robustness, ICLR 2020.

[6] Neural Attention Distillation: Erasing Backdoor Triggers from Deep Neural Networks, ICLR 2021.


**Summary Of The Paper:**

This paper proposes an iterative data filtering/expanding approach with a set of weak learners for backdoor defense. The core idea is that clean distribution and backdoor distribution is incompatible and thus making them seperable based on the expanding error of the training set with many smaller subsets. The weak learners refer to the snapshots of the classifier trained on different subsets and expanded sets.  It is an iterative approach with multiple (8) rounds of  ISPL (Inverse Self-Paced Learnin) and 24 weak learners (each trained for 40 epochs). The entire process is also repeated for 3 times. Theoretical formulation and jusstificaiton have been given along with some empirical verfifiction.

**Summary Of The Review:**

Good attempt, but the assumptions are too strong, making the entire theoretical analysis less convincing, problematic in several places, cherry-picked results,  missing many details about the overall training procedure, strategy for hyper-parameter choosing, missing experiments on many other attacks and datasets (only experiment with the simplest BadNets attack and CIFAR-10 datasets is given). Overall, I don't believe the proposed defense approach is good or simple enough to be useful in practice.

---

> ### Author Response · Authors · 2021-11-29
> **Responses (1/2)**
>
> Thanks very much for the detailed questions. Our responses are below
>
> >    The threat model adopted in this work needs more justifications. Backdoor attacks are no longer purely data poisoning anymore, there are quite a number of them are “model poisoning” attacks that follow a different setting. They follow different threat models in terms of who controls the training process. I understand this paper follows the more traditional data poisoning threat model, but it needs to be made very clear and contrasted to other backdoor defense threat models.
>
> We will expand the related works section to include a discussion of other attack models. We selected the standard backdoor model since it is the easiest to analyze and our evaluation indicates that even the relatively standard backdoor attack remains to be solved.
>
> >    Where the authors place their defense in the current literature is not clear. It appears to me that the proposed method is a robust training method, not a detection or backdoor removal method. The defense scope should also be discussed explicitly. The proposed method can only defense dirty-label backdoors with simple trigger patterns.
>
> Thanks, we will clarify this. Note that a large number of defenses also require similar access to the training set for retraining (e.g., all the defenses we use for comparison).
>
> >    Some of the assumptions are too strong, and their implications to backdoor attack/defense should also be explained. Property 4.4 (Incompatibility of primary and noise distributions), this assumption e most excludes most advanced backdoor attacks (trojan [1], dynamic input-aware [2], sample-wise [3], clean-label [4], feature-space attack, invisible attack, blending attack, reflection attack, and many others), and limits it to only simple attacks like BadNets. The authors can find many other examples in this GitHub repo: https://github.com/THUYimingLi/backdoor-learning-resources. Backdoor examples are by design compatible to clean data distribution, iconic characteristic of backdoor attacks to traditional data poisoning attacks. The tested BadNets is a classific dirty-label backdoor attack, which is very close to the traditional poisoning attacks but can no longer represent all backdoor attacks.
>
> Again, we note that existing defenses are broken by the classic dirty-label backdoor attack, as evidenced by our evaluation, particularly against the strong single-pixel trigger.
>
> >    Theorem 4.5 (Sets minimizing expansion error are homogeneous) is based on the strong assumption of incompatibility. I doubt it will hold for backdoor attacks, not just because of the incompatibility assumption, but also in general without knowing the poisoning rate. The authors can prove me wrong with 40% poisoning but keeping the same number of samples. How to calculate the number of remaining/removed samples anyway?
>
> As stated our theory gives consistent results so long as one is willing to rerun the defense for a very large amount of time (see the stochastic version of Theorem 4.1). Our results indicate that one can expect good defense within an acceptable run time for up to 20% poison--which we believe is extremely high in practice.
>
> >    How does Algorithm 1 help is not well explained. The input has homogenous sets S_1-S_N, the return is their votes (based on their mean empirical error), is this to choose low-loss samples like in ITLM, Shen and Sanghavi, 2019. How does this differ from their strategy except using the ensemble (but in the full set case)? How the votes contribute to Algorithm 2?
>
> Algorithm 1 (boosting) is given homogeneous sets, i.e., each set is either entirely poison or entirely clean. We still need some way to identify which set are clean. Algorithm 1 produces such an identification (more specifically, it produces a partition of sets into a clean partition and a poison partition--then assuming there are more clean data points than poisoned data points, we can recover the clean data).
>
> Note that Algorithm 2 (ISPL) is the input to Algorithm 1 (boosting), not the other way around.
>
> > In Proof of Theorem 4.5, , why?
>
> This is notation to simplify the exposition, please see the top of page 13.
>
> > What is the overall training procedure? How time-consuming it is compared to standard training. The training details given in the appendix show that the proposed method is complex, ad hoc, and hard to tune for other datasets, attacks, or poisoning rates.
>
> Each partitioning is roughly equivalent to training end-to-end once. Since we partition three times, our approach takes roughly 4x the amount of time compared to regular training. (Note that the partitions are independent and can be performed in parallel.)
>
> Our ablation studies indicate that our method is fairly robust to the choices of the main hyperparameters alpha and beta (Appendix C.2)

---

> > ### Author Response · Authors · 2021-11-29
> > **Responses (2/2)**
> >
> > > Confusing notations: expansion/compatibility, -compatibility, and the purposes of the samples determined by and in Algorithm 2.
> >
> > alpha always refers to compatibility. In Algorithm 2, beta refers to the fixed size of the set selected as defined in (13). We will change the notation in Assumption 4.3 to remove the additional usage of beta (thanks!)
> >
> > > The biggest concern about the experiment, except for missing many different types of attacks and datasets, is the cherry-picking results in the main text, Table 1. The complete results in appendix Table 3-5 indicate many failure cases on some of the source-target classes, for example, 3/5, 7/4, while the results in Table 1 only show the successful ones. So, based on the more complete results, the proposed defense does work at least on some source-traget class pairs. I doubt it will work on high poisoning rate or other types of advanced attacks like input-aware, sample-wise, clean labels attacks.
> >
> > Please note that the attack is very strong in that none of the existing baselines successfully defend in any meaningful way, which is consistent with the results presented in Table 1.
> >
> > Additionally, as part of a funded research program, we have submitted our defense to a third-party for evaluation on the German Traffic Sign Recognition Benchmark [1] with the following results:
> >
> > | poison % | clean acc % (higher is better) | poison misclass % (lower is better) |
> > |----------|--------------------------------|-------------------------------------|
> > | 0        | 92.6                           | x                                   |
> > | 1        | 91.8                           | 2.5                                 |
> > | 5        | 92.4                           | 2.2                                 |
> > | 10       | 92.2                           | 2.9                                 |
> >
> > > Missing discussion of other defense methods: detection methods and backdoor removal methods. Can detection methods be directly applied to detect and remove the backdoor examples, if one can retrain the model on the purified data? Can backdoor trigger removal methods like Model be used or compared? What's wrong with the assumption of having a small set of clean samples, if the authors assume the defender has access to the full dataset and the dataset is less than 50% poisoned?
> >
> > These are interesting questions, but generally outside of the scope of this work. Again, we emphasize that assuming access only to a potentially poisoned dataset and the ability to train is a very common setting [2,3,4].
> >
> > > The authors kept arguing that many existing defense works are not tested on CIFAR-10, why the authors didn’t test other datasets since many of them did? What makes the CIFAR-10 dataset so special? Transferability of the developed defense to large datasets? CIFAR-10 is not a good backdoor dataset is simply because its resolution is low, which is not an ideal place to design and test different types of trigger patterns, and even they work on CIFAR-10, they may not transfer to real-world scenarios which are surely not low resolution.
> >
> > Please see above, GTSRB is a higher resolution dataset with 43 classes of substantial practical interest. Most of the works which do not test on CIFAR-10 use *simpler* datasets (in the sense that one can achieve higher clean accuracy, such as Celeb-A or MNIST).
> >
> > We also find that apples-to-apples comparisons of performance is very difficult. Taking an example defense from the reviewer's comments, Neural Attention Distallation [5] published at ICLR 2021 tests only on CIFAR-10 and GTSRB. However, their models achieve only 85% clean accuracy on CIFAR-10 before defense, and only 81% clean accuracy afterward, compared to our work which uses models achieving 94-95% clean accuracy before defense, and 92-93% clean accuracy afterward. Additionally, for the BadNets attack, they use only the large 3x3 grid trigger, and test only a single target label (0). Hence it is hard to say whether the defense should generalize to models trained to higher accuracy or other target labels / triggers.
> >
> > [1] Stallkamp, Johannes & Schlipsing, Marc & Salmen, Jan & Igel, Christian. (2011). The German Traffic Sign Recognition Benchmark: A multi-class classification competition. Proceedings of the International Joint Conference on Neural Networks. 1453 - 1460. 10.1109/IJCNN.2011.6033395.
> >
> > [2] Brandon Tran, Jerry Li, and Aleksander Madry. Spectral signatures in backdoor attacks. arXiv
> > preprint arXiv:1811.00636, 2018.
> >
> > [3] Bryant Chen, Wilka Carvalho, Nathalie Baracaldo, Heiko Ludwig, Benjamin Edwards, Taesung
> > Lee, Ian Molloy, and Biplav Srivastava. Detecting backdoor attacks on deep neural networks by
> > activation clustering. arXiv preprint arXiv:1811.03728, 2018.
> >
> > [4] Yanyao Shen and Sujay Sanghavi. Learning with bad training data via iterative trimmed loss
> > minimization. arXiv preprint arXiv:1810.11874, 2019.
> >
> > [5] Neural Attention Distillation: Erasing Backdoor Triggers from Deep Neural Networks, ICLR 2021.

---

### Official Review · Reviewer_gSj9 · 2021-10-31

**Correctness:** 3
**Technical Novelty And Significance:** 3
**Empirical Novelty And Significance:** 3
**Recommendation:** 5
**Confidence:** 3

**Main Review:**

Strengths:
- The authors tackle a very relevant problem
- The proposed solution is creative
- There are strong theoretical motivations and modeling

Weaknesses:
- The empirical evaluation is only on one dataset (CIFAR-10) - I wonder whether this overall approach may include an unwanted smoothing of decision boundaries
- Missing defenses (e.g., MNTD [2])
- No adaptive adversary is considered [1]
- Intuitions are not always clear before diving into the details; it would be useful also to draw a high-level pipeline of the proposed method for clarity

Comments:

The paper proposes an idea which is strongly formalized and thoroughly theoretically thought.

I do have some concerns regarding the general efficacy of the approach:
- **Missing adaptive adversary**: When proposing a new defense to adversarial attacks, it is fundamental to consider an attacker that knows which defense you are employing [1]. In the empirical evaluation, you only consider other proposed defenses, but do not consider a backdoor attack variant which knows you are looking for these homogeneous sets and then using the ensemble of weak learners. I feel you may be hence overestimating the effectiveness of your defense.
- **Missing relevant defense**: At S&P21 it was recently proposed a new backdoor detection mechanism called MNTD [2], which has shown to be extremely effective in detecting backdoored models. I think the authors should also consider and compare against this other defense. But, more importantly, I feel their defense should be evaluated against an adaptive attacker as mentioned in the previous point.
- **One dataset**. The empirical evaluation is conducted only on CIFAR-10. So I wonder if there is any bias in this. For example, does your approach involve a smoothing of the boundaries? What would happen if you use FMNIST or EMNIST? Would the clean model accuracy still hold? I have the suspicion that your approach may be implicitly increase the smoothing of the decision boundaries to improve generalization, and I wonder what effects this may have on a larger ecosystem.
- **Intuitions and pipeline**. Although I strongly appreciate the heavy formalism of the paper, I also felt that having an overall bird-eye view of the pipeline would have helped in assessing its overall robustness. Hence, I feel that a diagram of two summarizing the intuitions behind the main phases of the proposed defense would greatly help the reader.
- **Minor comments**. Some other minor comments
	- some citations are missing and are marked with "(?)", such as MNIST and CIFAR-10.
	- Possible typo right before equation 11: extra or missing parenthesis


References:
- [1] Carlini, Nicholas, et al. "On evaluating adversarial robustness." arXiv preprint arXiv:1902.06705 (2019).
- [2] Xu, Xiaojun, et al. "Detecting AI trojans using meta neural analysis." 2021 IEEE Symposium on Security and Privacy (SP). IEEE, 2021.

**Summary Of The Paper:**

This paper proposes a new defense to backdoor attacks. In practice, it is an iterative training procedure which aims to remove poisoned data from the training set. This happens in two phases: an ensemble of weak learners identifies distinct homogeneous sub-populations in the training se, and a boosting framework aims to exclude poisoned data and recover clean data. They compare their approach with a few other defenses on CIFAR-10 and on dirty-label backdoor attacks.

**Summary Of The Review:**

I feel the major shortcoming is the lack of an adaptive adversary, possibly overestimating the efficacy of the overall defense and giving a false sense of security.

---

> ### Author Response · Authors · 2021-11-29
> **Responses**
>
> Thanks very much for the comments! We have addressed the major points below.
>
> >    Missing adaptive adversary: When proposing a new defense to adversarial attacks, it is fundamental to consider an attacker that knows which defense you are employing [1]. In the empirical evaluation, you only consider other proposed defenses, but do not consider a backdoor attack variant which knows you are looking for these homogeneous sets and then using the ensemble of weak learners. I feel you may be hence overestimating the effectiveness of your defense.
>
> Thanks for the comment. We think that the formalism does counteract this to the extent that the failure modes are clearly stated. [1] mainly addresses defenses that are primarily empirical in nature. Given the theory that supports our defense, which is a contribution of independent interest (particularly in the data poisoning field which is heavily dominated by empirical research on both the attacker and defender sides), we think designing an adaptive adversary is outside the scope of the present work.
>
> >    Missing relevant defense: At S&P21 it was recently proposed a new backdoor detection mechanism called MNTD [2], which has shown to be extremely effective in detecting backdoored models. I think the authors should also consider and compare against this other defense. But, more importantly, I feel their defense should be evaluated against an adaptive attacker as mentioned in the previous point.
>
> Thanks, we will be sure to include [2] in our discussion of related work. Note that their evaluation uses very large, non-standard triggers (see Table V) such as a heart outline which is practically the same size as the image. In comparison, we use more subtle triggers such as a single-pixel trigger, which in our evaluation is shown to break existing SOTA defenses. Additionally, we note that this paper proposes the defense on purely empirical grounds, in contrast to our work.
>
> >    One dataset. The empirical evaluation is conducted only on CIFAR-10. So I wonder if there is any bias in this. For example, does your approach involve a smoothing of the boundaries? What would happen if you use FMNIST or EMNIST? Would the clean model accuracy still hold? I have the suspicion that your approach may be implicitly increase the smoothing of the decision boundaries to improve generalization, and I wonder what effects this may have on a larger ecosystem.
>
> As part of a funded research program, we have submitted our defense to a third-party for evaluation on the German Traffic Sign Recognition Benchmark [3] with the following results:
>
> | poison % | clean acc % (higher is better) | poison misclass % (lower is better) |
> |----------|--------------------------------|-------------------------------------|
> | 0        | 92.6                           | x                                   |
> | 1        | 91.8                           | 2.5                                 |
> | 5        | 92.4                           | 2.2                                 |
> | 10       | 92.2                           | 2.9                                 |
>
> >    Intuitions and pipeline. Although I strongly appreciate the heavy formalism of the paper, I also felt that having an overall bird-eye view of the pipeline would have helped in assessing its overall robustness. Hence, I feel that a diagram of two summarizing the intuitions behind the main phases of the proposed defense would greatly help the reader.
>
> Thanks for the suggestion! We will prepare a revised version with a high level diagram that provides an overview of our approach.
>
> References:
>
> [1] Carlini, Nicholas, et al. "On evaluating adversarial robustness." arXiv preprint arXiv:1902.06705 (2019).
>
> [2] Xu, Xiaojun, et al. "Detecting AI trojans using meta neural analysis." 2021 IEEE Symposium on Security and Privacy (SP). IEEE, 2021.
>
> [3] Stallkamp, Johannes & Schlipsing, Marc & Salmen, Jan & Igel, Christian. (2011). The German Traffic Sign Recognition Benchmark: A multi-class classification competition. Proceedings of the International Joint Conference on Neural Networks. 1453 - 1460. 10.1109/IJCNN.2011.6033395.

---

### Official Review · Reviewer_ey67 · 2021-10-31

**Correctness:** 4
**Technical Novelty And Significance:** 4
**Empirical Novelty And Significance:** 4
**Recommendation:** 8
**Confidence:** 3

**Main Review:**

Pros:
1. I like the description of self-expansion and compatibility in Section 4.1. It is very clear and properly motivates the idea of homogeneous set. The math proof of Theorem 4.5 makes sense to me (did not check every detail).
2. A more challenging attack case (1 pixel trigger for CIFAR-10) is used in the experiments to validate the effectiveness of proposed defense. It significantly outperforms existing work without hurting clean accuracy too much.

Cons:
I did not see obvious weakness of this paper. I am interested to see a running complexity comparison between previous proposed work. How does the number of ISPL runs affect the defense performance?

Typos:
1. Overview last paragraph: CIFAR-10 (?)
2. Background and setting first paragraph: a loss function over Y X Y -> a loss function over X X Y

**Summary Of The Paper:**

This paper tackles backdoor attacks where an adversary performs targeted attack against neural networks by injecting some poisoned data into the training data without sacrificing prediction accuracy on clean data. The defense works by recovering the clean data from poisoned training data. The authors proposed an Inverse Self-Paced Learning (ISPL) algorithm to first find a set of homogeneous sets (pure clear or poisoned data) and use an ensemble of weaker learners to exclude poisoned data from training set.

**Summary Of The Review:**

I will vote for accepting this paper. The idea of separating training set into self-expanding (homogeneous) sets is novel and interesting. The use of a set of weak learners and proposed boosting algorithm are technically sound. The experimental part demonstrates the performance of proposed defense over other existing work.

---

> ### Author Response · Authors · 2021-11-29
> **Response**
>
> Thanks very much for the supportive comments! We have fixed the typos and will add an additional study ablating the number of runs of ISPL.

---

### Official Review · Reviewer_21eY · 2021-11-05

**Correctness:** 3
**Technical Novelty And Significance:** 2
**Empirical Novelty And Significance:** 1
**Recommendation:** 3
**Confidence:** 4

**Main Review:**

The goal of the paper is to provide a defense against backdoor attacks. They define backdoor attacks in a very specific way: We have a distribution $D$, and a "backdoored" version $D'$ that is, equal to $(1-p)\cdot D+ p\cdot backdoor(D)$ for some poisoning ratio $p$. And the $backdoor()$ function is a function that takes a labeled example $(x,y)$ and outputs $(x',y')$ where both $x'$ and $y'$ are deterministic functions of $x$ and $y$. The backdoor function should be chosen by adversary a priori. Then the success of the attack is measured by the accuracy of the resulting model on distribution $backdoor(D)$.

After defining this threat model, they try to provide an outlier detection technique that when given a set $S$ sampled from $D'$ can separate the backdoored samples from benign samples. If I understand correctly, in the rest of paper, they try to find a way to solve this problem.

They define a notion of self-expansion of a set $S$ with respect to another set $T$, and with a ratio alpha. This notion is intuitively the expected empirical risk of a classifier trained on a dataset that is the union random subset of S with size $\alpha|S|$ and $T$. And the empirical risk is calculated on the set $S$. This notion captures how well a random subset of S generalizes to the rest of the set, when combined with $T$.

Then define the notion of $\alpha$-compatibility of two sets $S$ and $T$ that is defined based on whether the self-expansion of set $S$ with respect to $T$ is smaller than without $S$ (compatible) or not (not compatible).

From here on, they start making assumption that helps them to identify the outliers. The first assumption is that the self-expansion is a convex function of alpha, for both inliers and outliers. This assumption is extremely strong. The second assumption is about monotonicity of compatibility. That is, if $S$ and $T$ are $\alpha$-compatible, then they should be $\beta$-compatible too, for all $\beta>\alpha$. This assumption is again an strong assumption.

They also need some other assumptions that any subset of the outliers is incompatible with the incliners. This assumption is actually something that I think cannot hold in any setting and I mention that below in my comment to authors.

Then using all these strong assumptions, the authors show that they can separate outliers from inliers using some tests for compatibility.



Questions to Authors:

1- To me, this paper is mostly about outlier detection. It is not clear to me why authors focus on backdoor poisoning attacks in the introduction as rest of the paper does not have anything to do with backdoor attacks.

2- I have strong concerns about property 4.4. How can we strict and complete incompatibility? If you consider the empty set as a subset of T, then strict and complete incompatibility is not possible.

3- What is the role of alpha of property 4.4 in the theorem 4.8?

4-In general, there is a need for more discussion about why the assumptions are realistic and "mild" (as stated in the paper). In my opinion, these assumption are really strong. Also, the adversary can easily circumvent these assumptions. Why should we only consider adversaries that adhere to these assumptions?

5-Why do you need to define multiple primary and multiple noise distributions? In my opinion this only makes the paper harder to understand.



Other comments:

-"Clearly, given enough training samples S = {(x1, y1)...,(xn, yn)} iid from D, the empirical risk gets arbitrarily close to the population risk." This sentence is clearly wrong unless you assume something about your function class.

-Theorem 4.5: \{(\alpha_i,\mathcal{D}_i\}-> \{(\alpha_i,\mathcal{D}_i)\}

-Page 1 and 8: citation for CIFAR is missing

-Page 3, eq 6: Why should \pi be a  permutation? Could it be any mapping? Or does it even need to be a mapping between labels? Can't the adversary sometimes label a cat as a dog and sometimes label a cat as a frog?

-Theorem 4.8: What is delta?

-It's not clear how the learning objective is related to the threat mode.



**Summary Of The Paper:**

This paper studies a defense against poisoning attacks using an outlier detection technique they develop using a notion of "self-expanding" sets. They assume a number of properties for inliers and outliers and can classify them so long as those properties are satisfied. They run experiments based on an existing backdoor attack and they show that their technique could be successful.



**Summary Of The Review:**

This paper try to provide a defense against backdoor attacks through outlier detection. I find the direction of providing provable guarantees for outlier detection is interesting. However, the assumptions in the paper are very strong which makes the threat model of the paper narrow. These assumptions are not just about the data distribution, rather, they are about the properties of the distribution generated by adversary that can be easily violated by adversary.

I find the presentation of paper a bit confusing. The introduction talks about backdoor attacks but the technical parts of the paper has nothing to do with backdoor attacks.

I also have concerns about some of technical steps of the proof that I mentioned in my comments. I also have a hard time understanding the main theorem that there seems to be an undefined term (delta). Also, I don't see any connection to alpha in alpha-compatibility, which is one of the assumptions of the theorem.

---

> ### Author Response · Authors · 2021-11-29
> **Responses**
>
> We thank the reviewer for the detailed comments. Our responses are below.
>
> > 1- To me, this paper is mostly about outlier detection. It is not clear to me why authors focus on backdoor poisoning attacks in the introduction as rest of the paper does not have anything to do with backdoor attacks.
>
> The main distinguishing feature of backdoor attacks that we focus on is that training on the poisoned training set achieves the same accuracy on the clean test set as training on the clean training set. In other words, introducing poison does not affect clean accuracy (at the dataset level). Note that this is a very strong attack, since we essentially assume that the attack is *completely undetectable* during regular testing. The self-expansion property is specifically targeted toward counteracting this strong property. While other types of outliers may exhibit similar characteristics, they are not the focus of this work.
>
> > 2- I have strong concerns about property 4.4. How can we strict and complete incompatibility? If you consider the empty set as a subset of T, then strict and complete incompatibility is not possible.
>
> Note that in definition 4.2 we have specified non-empty subsets. (See also response to 4 below.)
>
> > 3- What is the role of alpha of property 4.4 in the theorem 4.8?
>
> alpha is the subsampling rate. In this work, we assume that alpha satisfying property 4.4 is known a priori and show how to leverage it to separate the clean data from the poisoned data. Our ablation studies indicate the performance is fairly robust to the choice of alpha in practice (Appendix C.2, Table 5)
>
> > 4-In general, there is a need for more discussion about why the assumptions are realistic and "mild" (as stated in the paper). In my opinion, these assumption are really strong. Also, the adversary can easily circumvent these assumptions. Why should we only consider adversaries that adhere to these assumptions?
>
> We do not claim to defend against adversaries that violate these assumptions. Our evaluations indicate that at least one class of strong adversaries (hidden trigger backdoor) satisfies these assumptions, or comes close enough for our defense to succeed where previous SOTA defenses fail.
>
> We also believe it is a substantial strength that we have a defense that depends on clearly stated and intuitive assumptions, even if they are somewhat idealized--i.e., our defense is not a case of "security by obscurity". Many defenses in this space are heuristic and it is not clear how or why they should work, or indeed, when they would fail--which in our opinion is a major defect.
>
> > 5-Why do you need to define multiple primary and multiple noise distributions? In my opinion this only makes the paper harder to understand.
>
> The additional generality shows that our approach applies even when there are sub distributions (e.g., multiple classes) which is often a challenge for, e.g., clustering-type approaches which have high false positive rate (see for instance Activation Clustering [1] in Appendix C, Tables 3 and 4).
>
> > Other comments:
>
> > -"Clearly, given enough training samples S = {(x1, y1)...,(xn, yn)} iid from D, the empirical risk gets arbitrarily close to the population risk." This sentence is clearly wrong unless you assume something about your function class.
>
> This is referring to the convergence of the risk estimate for an individual classifier (via LLN), not the convergence of the empirical risk minimizer to the true risk minimizer. We will clarify this statement.
>
> > -Theorem 4.5: {(\alpha_i,\mathcal{D}_i}-> {(\alpha_i,\mathcal{D}_i)}
>
> > -Page 1 and 8: citation for CIFAR is missing
>
> Thanks very much! These have been addressed.
>
> > -Page 3, eq 6: Why should \pi be a permutation? Could it be any mapping? Or does it even need to be a mapping between labels? Can't the adversary sometimes label a cat as a dog and sometimes label a cat as a frog?
>
> Yes, this should be possible. To simplify the exposition, our threat model is not meant to be fully general, only general enough to capture the standard hidden trigger backdoor attack setting so that we can precisely define the targeted misclassification risk (7).
>
> > -Theorem 4.8: What is delta?
>
> delta is the gap in the incompatibility (Definition 4.2). We will clarify this, thanks!
>
> > -It's not clear how the learning objective is related to the threat mode.
>
> Our learning objective is to (1) minimize the targeted misclassification risk while also (2) maintaining a high clean accuracy. We need the threat model to define the targeted misclassification risk. Note that this contrasts with simply maintaining a high clean accuracy in the presence of poison, which is not sufficient to evaluate a defense against backdoor attacks.
>
> [1] Bryant Chen, Wilka Carvalho, Nathalie Baracaldo, Heiko Ludwig, Benjamin Edwards, Taesung Lee, Ian Molloy, and Biplav Srivastava. Detecting backdoor attacks on deep neural networks by activation clustering. arXiv preprint arXiv:1811.03728, 2018.

---

### Decision · Program_Chairs · 2022-01-20

**Decision:**

Reject

**Comment:**

The paper presents a new defense against backdoor attacks based on the discovery of homogeneous populations in the training data and subsequent filtering of poisoned data due to its difference from the said populations. The method has a solid theoretical foundation which, however, requires strong assumptions on attacks and benign data. Due to these assumptions the theoretical guarantees alone cannot ensure that the defense is robust against adaptive attacks. The experimental validation of the proposed method is limited to one benchmark datasets (CIFAR), additional results are briefly presented in the response but not elaborated on.